# Surface Properties and Tribological Behavior of Additively Manufactured Components: A Systematic Review

Christian Orgeldinger *, Armin Seynstahl, Tobias Rosnitschek and Stephan Tremmel

Engineering Design and CAD, University of Bayreuth, Universitätsstraße 30, 95447 Bayreuth, Germany; armin.seynstahl@uni-bayreuth.de (A.S.); tobias.rosnitschek@uni-bayreuth.de (T.R.); stephan.tremmel@uni-bayreuth.de (S.T.)
* Correspondence: christian.orgeldinger@uni-bayreuth.de

**Abstract:** Innovative additive manufacturing processes for resilient and sustainable production will become even more important in the upcoming years. Due to the targeted and flexible use of materials, additive manufacturing allows for conserving resources and lightweight design enabling energy-efficient systems. While additive manufacturing processes were used in the past several decades mainly for high-priced individualized components and prototypes, the focus is now increasingly shifting to near-net-shape series production and the production of spare parts, whereby surface properties and the tribological behavior of the manufactured parts is becoming more and more important. Therefore, the present review provides a comprehensive overview of research in tribology to date in the field of additively manufactured components. Basic research still remains the main focus of the analyzed 165 papers. However, due to the potential of additive manufacturing processes in the area of individualized components, a certain trend toward medical technology applications can be identified for the moment. Regarding materials, the focus of previous studies has been on metals, with stainless steel and titanium alloys being the most frequently investigated materials. On the processing side, powder bed processes are mainly used. Based on the present literature research, the expected future trends in the field of tribology of additively manufactured components can be identified. In addition to further basic research, these include, above all, aspects of process optimization, function integration, coating, and post-treatment of the surfaces.

**Keywords:** additive manufacturing; 3D printing; lightweight design; surface; tribology





## 1. Introduction

The necessity to minimize used resources and reduce energy consumption drives today's product development, with an overreaching goal toward an industrial green transition. A significant share of global energy consumption is attributed to loss due to wear and friction [1], which reflects the importance of tribology in nearly every sector of technical products. Accordingly, research on minimizing friction is inevitable within the context of a green transition, thus sounding the bell for the golden age of tribology [2].

Furthermore, to reduce resource consumption, in the first place, structural optimization tools are common practice to determine the best material distribution within a given design space under predefined loads, leading to bioinspired designs, which have yet to be materialized in the physical world [3,4]. Additive manufacturing (AM) provides this needed capability to directly manufacture digital parts that are decentralized and all over the world; therefore, it can be also considered the physical arm of digitalization. Its objective under the umbrella of a green transition is to create optimized products with minimized material effort. Since its development in the early 1980s, AM evolved from the rapid prototyping of visual aids to the direct manufacturing of complex end-use parts [5,6]. With the recent but continuously increasing additively manufactured parts used in technical systems, questions about their tribological behavior are on the rise as well. Thus, the understanding

of synergies between tribology and AM can pave the way for novel solutions to address future techno-societal challenges.

To pave the way for a better understanding of interactions and synergies of tribology and AM, a more detailed analysis of the used AM techniques, materials, and fields of application is essential. As other reviews focus solely on metals [7], or laser-based AM methods [8], a global and systematic overview is missing. Accordingly, this contribution summarizes the current trends of materials and applications in AM with respect to tribology. Thereby, this contribution differentiates between studies primarily focusing on the part's surface and studies with a distinct link to tribological behavior. Within this context, the following questions are addressed in particular:

- Which materials have already been investigated?
- Which AM machines were mostly used?
- Is the conducted work related to basic research or specific applications?

Thus, the interested reader shall be provided with a high-level understanding of the capabilities of current developments in AM for specific materials with respect to tribological applications.

## 2. Methodology

The systematic literature survey followed the updated Prisma statement [9] and used the Web of Knowledge database (Clarivate Analytics, London, UK). The search was restricted to the English language only, and the time span was between January 2001 and September 2022. During the search, the following queries were used:

- Tribology and additive manufacturing;
- Friction and additive manufacturing;
- Wear and additive manufacturing;
- Surface roughness and additive manufacturing;
- Surface characterization and additive manufacturing;
- Bio-tribology and additive manufacturing.

Where applicable, British and American English spelling variants of the queries were used. The PRISMA flowchart of the systematic protocol for the article collection is presented in Figure 1a.

The excluded records were not in the English language or out of scope (for instance, friction stir welding). In addition, articles without a digital origin identifier (DOI) were not considered in this review. Starting from 2011, a total of 293 records were identified, from which 167 were included in this synthesis. The number of records slowly increased between 2011 and 2017. Between 2001 and 2010, no contributions were found that met the search criteria. Since 2018, the number of records rapidly increased and doubled in the last year, as shown in Figure 1b. Accordingly, 36.9% of the analyzed articles were published between January and September 2022. To set these numbers into a wider frame, namely the topic "tribology" in general, publications per year averaged about 563 between 2011 and 2019 and nearly tripled, to 1431 publications per year, between 2020 and 2022. For the publications included in this study, an average of 5.5 articles per year were published between 2011 and 2019, which increased by a factor of nine to 46 publications between 2020 and 2022. Accordingly, the research activities revolving around the topic "tribology and AM" increased significantly stronger as in "tribology" in general.

This enormous growth of interest can be attributed to the emerging maturity of manufacturing processes, which allow for relatively robust products suitable for serial parts. In addition, material costs in general significantly increased in 2022, which motivated researchers to investigate the use of additively manufactured lightweight components to save costs. Furthermore, AM strengthens companies' resilience and enables flexible adaptions to political or societal crises in general. Hence, these three pillars foster the necessity to assess the tribological behavior of additively manufactured parts that can be

clearly aligned with the doubling of published research between 2021 and the first three quartiles of 2022.

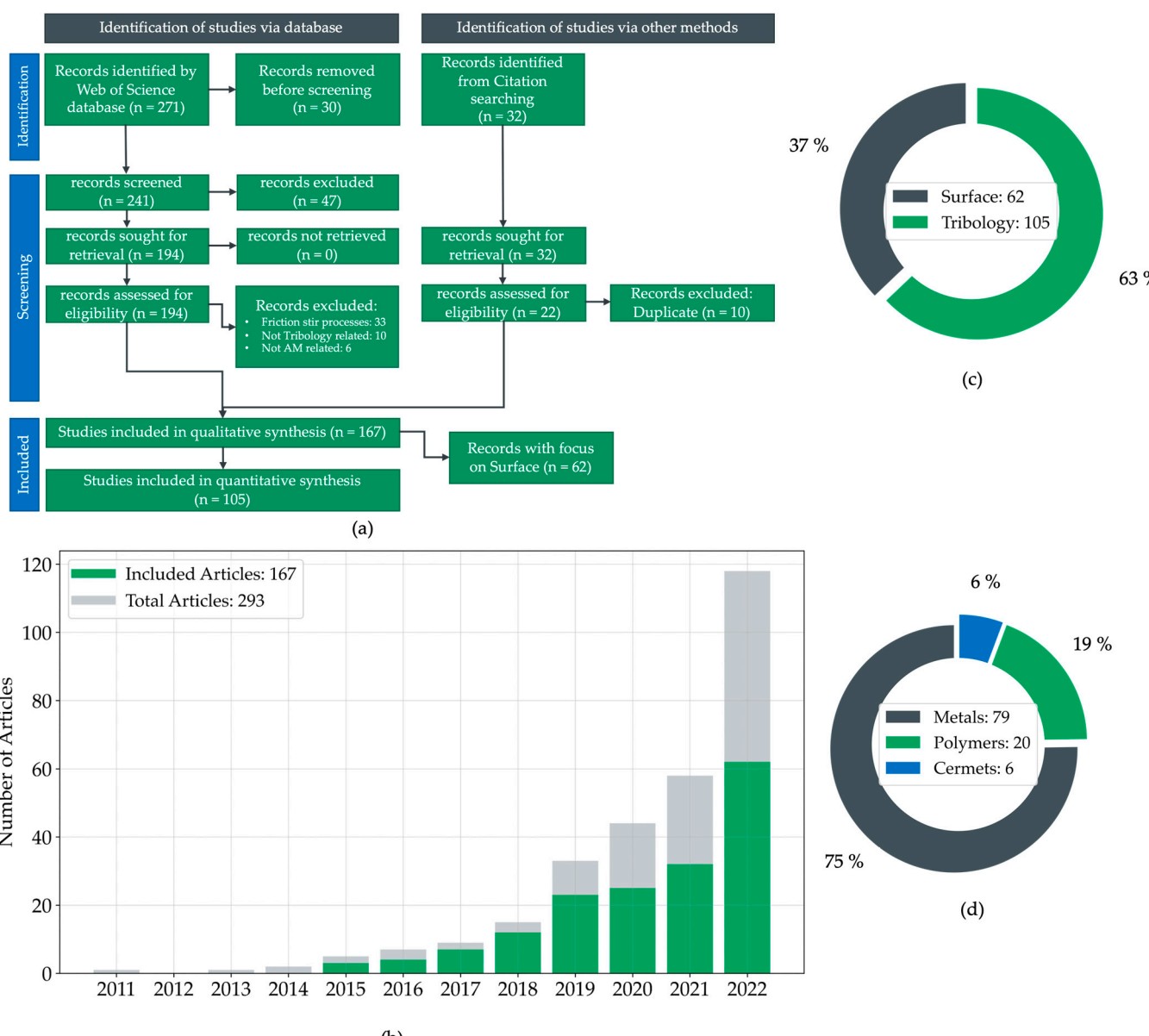

**Figure 1.** PRISMA protocol of the conducted systematic search (**a**) and number of publications per year (**b**), clustered based on main categories (**c**), and publications assigned to tribology clustered according to the material of the AM component (**d**).

For qualitative and quantitative analyses, the included articles were further divided into two main categories: "surface" and "tribology" (Figure 1c).

Within the surface category, all the clustered articles primarily deal with the surface properties of additively manufactured parts, without a distinct link to tribological behavior. Since this is not the key aspect of the recent review, this category was restricted to qualitative synthesis only. The remaining 105 articles were subjected to quantitative synthesis and clustered based on the materials used, which were divided into the subcategories of polymers, metals, and ceramics/cermets, as presented in Figure 1d.

## 3. Surface

This first section provides a precise overview of research dealing with the surface of additively manufactured parts, with no primary link to tribological behavior. Accordingly, the objectives of the reviewed research papers focused on surface quality and were clustered into five categories (Figure 2).

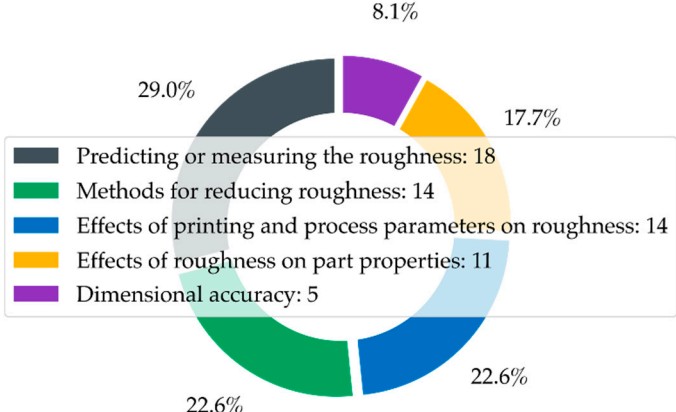

**Figure 2.** Publications ($n = 62$) attributed to the surface category clustered into five categories.

In total, 62 articles were identified for the qualitative analysis, within which 18 articles belong to the category **predicting or measuring the roughness**. In particular, the non-destructive measurement or modeling, simulation, and prediction of roughness can be key aspects of quality assurance and help to leverage the full potential of additive manufacturing. Therefore, 10 articles dealing with metal additive manufacturing were identified. Interestingly, an equal number of articles dealing with the relatively new wire-arc additive manufacturing (WAAM) [10–14] and more conventional laser powder bed fusion processes for metal (LPBF-M) [15–19] could be identified. From the analyzed research, the use of machine learning in roughness prediction should be highlighted. Lu and Shi [16] used a design-of-experiment approach with a central composite design for predicting surface roughness in LPBF-M, which led to an accuracy $R^2$ of 74.36%. Furthermore, Kumar and Jain [10] deployed a $k$-nearest neighbor algorithm in WAAM with a prediction error ranging from $-5.8\%$ to 2.3%. Additionally, Xia et al. [12] investigated various machine learning models for predicting WAAM surface roughness, from which adaptive neuro-fuzzy inference systems optimized with the genetic algorithm showed superiority, with an $R^2$ of 93.52%.

Furthermore, six articles investigated polymer parts, the majority of which belong to fused filament fabrication (FFF) processes [20–26], and one article investigated PVA composites in selective laser sintering (SLS) processes [27].

The next category gathers **methods for reducing the roughness of additively manufactured parts**, where 14 articles were identified. From those, solely the work of Iquebal et al. [28] deals with polymer parts, while metal parts are the subjects of the remaining articles. Considering metals, a reduction in roughness is predominantly achieved using post-processing steps. For instance, Jiang et al. [29] investigated electropolishing on the nickel-based superalloy Hastelloy® X and achieved a reduction from 10.3 µm as-built surface roughness down to 1.2 µm.

Furthermore, all the articles dealing with the **effect of printing and process parameters on roughness** were gathered into another category. In this category, eleven articles focusing on metal AM processes [30–40] and three articles, which focus on polymer parts in FFF [41–43] were identified. For instance, Panahizadeh et al. [35] investigated laser powder bed fusion and optimized the parameters of laser power, scanning speed, hatch space, scanning pattern angle, and heat treatment temperatures, using the multiobjective non-dominant sorting genetic algorithm, to obtain parameter combinations that lead to maximum relative density and minimum roughness on Ti6Al4V samples. Mushtaq

et al. [43] used the Taguchi method to explore the influence of fused filament fabrication parameters on the roughness of printed ABS and PA on six specimens.

The **effect of surface roughness on part properties** was also the subject of several studies focusing on mechanical properties, such as microhardness [44], fatigue strength [45–48], corrosion resistance [49,50], and electrical properties of the parts [51].

As the last category, **dimensional accuracy** should be noted, which is the only category where studies on polymer AM [52–55] outweighs metal AM [54].

Derived from the investigated studies, it can be concluded that there is already a certain awareness of the influence of roughness on multiple part properties; furthermore, a common agreement exists that the roughness of additively manufactured parts must be improved for the use of end parts. Accordingly, it is obvious that most studies deal with either predicting or measuring the roughness in additive manufacturing, followed by studies that investigated methods for reducing the roughness.

All the investigated studies are summarized in Table 1 to provide a quick overview of the research on the roughness of additively manufactured parts, which is not primarily linked to the field of tribology.

**Table 1.** Overview of research for roughness in AM not primarily linked to the field of tribology.

| Category | Topic | Manufacturing Process | References |
|---|---|---|---|
| Predicting or measuring the roughness | Metal | LPBF, WAAM | [10–19] |
| | Polymer | SLS, FFF | [20–27] |
| Methods for reducing roughness | Metal | LBF, DMLS | [29,50,56–66] |
| | Polymer | | [28] |
| Effects of printing and process parameters on roughness | Metal | LBF, WAAM, GMAW | [30–40] |
| | Polymers | FFF | [41–43] |
| Effects of roughness on part properties | Corrosion | LPBF-M | [49,67] |
| | Mechanical | LPBF-M, Arbitrary | [44–48,51,68,69] |
| | Electric | FFF | [70] |
| Dimensional accuracy | Metal | LPBF-M | [54] |
| | Polymer | FFF, MJF, VP | [52–55] |

In the following sections, the current studies with a tribological context in the area of additively manufactured polymers, metals, and ceramics/cermets are summarized.

## 4. Tribology

### 4.1. Polymers

In the context of additive manufacturing, polymers play a very important role, due to their easy processability or low costs of manufacturing [71]. However, research carried out to date with regard to the tribological behavior of additively manufactured polymer parts is still quite scarce. Based on our literature review, only 20 of 103 publications deal with the tribological behavior of AM polymer specimens. The classification into subcategories was kept simple in this case, since only three different AM processes were investigated in the literature, which mainly differ according to the type of polymeric raw material and method of sample generation (Figure 3). The classification in subcategories is illustrated in Figure 3.

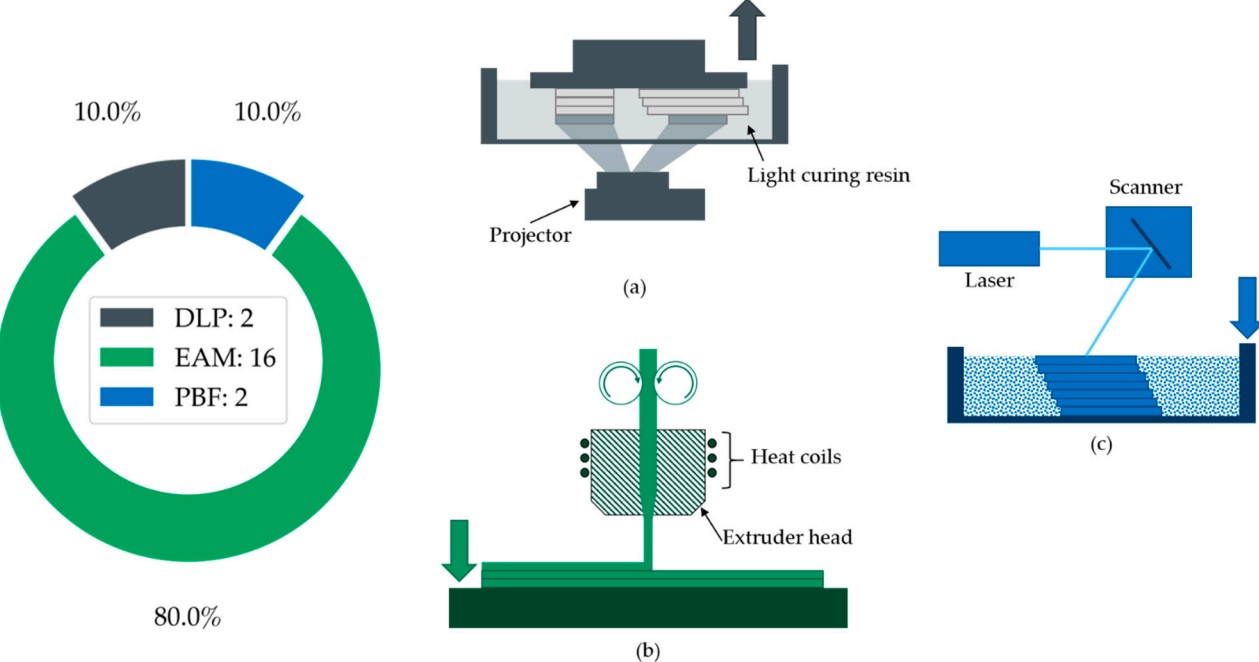

**Figure 3.** Classification in subcategories for polymers: (**a**) digital light processing (DLP); (**b**) extrusion-based additive manufacturing (EAM); (**c**) powder bed fusion (PBF).

**Extrusion-based additive manufacturing** (EAM) is the predominant technology (80% of polymer publications), followed by **powder bed fusion** (PBF) processes, in particular **selective laser sintering** (SLS), with 10% of publications, and two papers investigated polymer specimens using **digital light processing** (DLP). A further subdivision, for example, based on materials, tribological test methods, or applications, was not carried out due to the small number of publications. The publications considered in the following are summarized in Table 2.

### 4.1.1. Digital Light Processing (DLP)

Provided that the polymeric raw material is in a liquid form such as a resin, DLP technology allows the photopolymerization process to be initiated using irradiation with light, thus leading to the layer-by-layer build-up of specimens (Figure 3a).

Huettig et al. [72] used this process to print polymethylmethacrylate (PMMA) samples and compared them with conventionally milled parts in terms of wear behavior. Prior to the reciprocating wear tests, the sample surfaces were polished to a comparable roughness value ($R_a = 0.05 \pm 0.01$ μm). The reciprocating wear tests were performed under artificial saliva lubrication up to 5000 cycles at 5 N normal force and a sliding range of 10 mm. After evaluation, similar wear depths were measured for the differently manufactured specimens; thus, no significant difference in tribological behavior was detected. In a second study, Rudnik et al. [73] investigated the tribological behavior of the additively manufactured MED610 material for biomedical applications and demonstrated comparatively low abrasion resistance.

### 4.1.2. Powder Bed Fusion (PBF)

In powder-bed-based AM methods, the SLS process is predominantly used for polymers. Each layer is sintered with a defined height by using a laser beam, which produces the typical layer structure of AM components (Figure 3c).

To investigate the tribological behavior of SLS-printed samples, polyamide 12 (PA 12) [74,75] and thermoplastic polyurethane (TPU) [75] specimens were produced with different printing orientations: a horizontal printing direction corresponded to 0°, a perpendicular direction to 90°, and a diagonal direction to 45°, whereby all specifications

refer relative to the building platform and the powder bed, respectively. Ziegler et al. [75] showed that both COF and abrasive wear, evaluated using a reciprocating wear test, were lower for PA 12 than for TPU. This is possibly caused by the lower mechanical properties of TPU. A distinct printing orientation dependence was not discernible, whereas in [74], at least for PA 12, a considerably better tribological behavior with respect to friction and wear regarding the horizontal printing orientation (0°) was observed. However, hardly any difference was observed between the 45° and 90° directions in this case as well.

### 4.1.3. Extrusion-Based Additive Manufacturing (EAM)

EAM is the predominant technology for processing polymers, due to having a wide variety of feedstocks; inexpensive printers as opposed to, for example, powder-bed-based printers; and few requirements for safety aspects of processing [71,76]. In EAM technology, polymeric filament material is extruded through a nozzle to build up the components layer by layer (Figure 3b). This particular process is known as fused filament fabrication (FFF). For most feedstocks, the nozzle must be heated to decrease viscosity and improve extrusion. FFF was initially developed for polymeric materials, but it is feasible to extend the technology to other materials (metals, ceramics, etc.).

According to our literature review, considering the selected search parameters from Section 2, in the context of tribology and AM of polymers, the FFF process was the subject of most studies, with 16 out of 19 publications, as expected. Acrylonitrile butadiene styrene (ABS) [77–80] has emerged as an important feedstock, as well as composites with polycarbonate (PC) [81,82] and ABS reinforced with carbon fibers [83]. Furthermore, investigations have been carried out on polylactic acid (PLA) [79,80,84–86], which is very widely used in the AM industry, as a composite with bronze particles [87], silicon [84], and HT-PLA [86], which is suitable for higher temperatures. Finally, the following materials represented the research object of the considered publications and were investigated in detail with regard to their tribological behavior: polyethylene terephthalate glycol (PETG) [80,86,88], polybutylene terephthalate (PBT) [89], polyether sulfone (PES) [90], PES/BF (basalt fiber reinforced) [90], polyetherimide (PEI) [91], and acrylonitrile styrene acrylate (ASA) [92].

Starting with ABS, at least for prototyping or the production of the casing and operating parts, the possible suitability of ABS with regard to its use in tribotechnical systems is the subject of research. In their study, Amiruddin et al. [77] compared molded and FFF-printed ABS specimens on a ball-on-disc tribometer under paraffin oil lubrication and different loads. The authors observed an increase in the COF with increasing load, regardless of the manufacturing process. The FFF-printed components exhibited higher COF (0.040–0.055) due to higher surface roughness than molded ABS (0.009–0.025). In addition, higher loads resulted in decreasing wear rates. In [78], ABS test specimens were tribologically analyzed at different surface configurations, normal loads, and sliding speeds. The results showed a similar effect to [77], namely the COF increased with increasing normal loads, although the tribological tests were performed in a different test setup (reciprocating). Additionally, the COF decreased with higher sliding velocities. The authors also calculated and contrasted the results with theoretical tribological models, and it was found that these theoretical models cannot represent reality. This is mainly due to the fact that the FFF-printed surfaces are not smooth/polished surfaces [78].

Roy et al. [79] printed ABS specimens with different printing parameters, and particular consideration should be given to the infill percentage. After all, a lower infill percentage had a positive effect on the COF and resulted in its reduction. Besides model tests (pin on disc), in [80], printed ABS, PLA, and PETG polymeric gears were investigated for the component tests on a gear test rig under dry running against steel gears, to evaluate the wear behavior and service life. The tests indicated that PETG performed best in terms of both the wear behavior and service life, whereas the results of PLA and ABS were quite similar to each other and slightly below PETG. The tribological behavior of a composite material, which consists of modified ABS with PC, was investigated by Mohamed et al. in two publications using a non-lubricated pin-on-disc test, one in terms of friction [81] and



the other mainly related to wear behavior [82]. In both research papers, manufacturing recommendations were given by the tribological screening of samples manufactured with different printing parameters such as layer height, layer width, building direction, raster angle, or air gap. These recommendations contain the most optimal constellation of the investigated printing parameters with respect to the lowest wear rate and COF in the analyzed tribological system. Another possible modification of pure ABS involves carbon fiber reinforcement. Studies on the effect of tailored carbon fiber filler content on tribological behavior are provided in [83]. In this case, different printing parameters and carbon fiber filler content were investigated. According to the pin-on-disc tests, the results obtained indicate, on the one hand, an increasing COF at similar wear rates due to an increase in the filler content of carbon fibers under dry running and, on the other hand, a significant reduction in both COF and wear rates under water lubrication.

Several publications discuss the tribological behavior of both PLA and modified PLA. First of all, modifications of pure PLA by adding other materials such as bronze particles [87] or silicon (Si) [84] resulted in an improvement in mechanical properties, which in particular significantly improved the wear behavior. The reason is that the deformation of the substrate surface is reduced in sliding tests, which in turn increases the wear resistance and thus reduces the wear rate. In addition, a decreasing wear rate and COF were measured with increasing Si filler content regarding Si modification. In the cylinder-on-plate tests using the bronze–PLA composite material carried out without lubrication, the vertical printing orientation revealed the lowest wear rate in contrast to the horizontal printing direction, but the highest COF. Hanon et al. [86] investigated modified HT-PLA, which is more durable under high operating temperatures, and the melting point of the polymer increased. As in [87], the vertical printing orientation was found to be advantageous from a tribological point of view. In contrast to pure PLA and PETG, the results of HT-PLA exhibited the lowest friction and wear coefficients; hence, this polymer is preferred for tribological applications among the three polymers tested. As already mentioned, in [79], ABS specimens were compared with PLA in the same tribological system under the same test conditions, whereby PLA exhibited less friction but significantly higher material removal as a result of the observed wear processes. Another study investigated two different colored PLA filaments, with the aim of determining whether a difference in tribological behavior could be detected due to color [85]. The samples were printed at different temperatures, in natural and black colors. Tribological tests were performed on a pin-on-disc tribometer under dry conditions. The results of the tests indicated an influence of both filament color and printing temperature on tribological behavior. The black PLA showed a lower COF at the same printing temperature in contrast to the natural PLA, whereas 220 °C was the most favorable temperature for both variants.

Another publication, which deals solely with PETG without a direct comparison with other polymers, provides, for instance, the results on the influence of printing orientation or different loads under dry running conditions using a pin-on-disc tribometer [88]. The vertical printing direction, in contrast to the horizontal printing direction, as well as higher loads (independent of printing direction), led to a higher wear rate. In the case of frictional behavior, there was no direct correlation with the selected printing parameters. Nevertheless, a higher layer thickness resulted in a slightly higher coefficient of friction considering the same load conditions. The tribological behavior of PBT specimens was evaluated by screening different printing parameters, for example, infill percentage, infill variants, etc., on a pin-on-disc tribometer with reciprocating movement under dry conditions [89]. The aim was to identify a tribologically tailored printing variant. Concerning the wear rate, the printing variant with 90% infill degree, 90° infill orientation, 270 °C nozzle temperature, and a layer height of 0.25 mm performed most favorably. Vázquez Martínez et al. [92] investigated the tribological behavior of ASA under similar conditions. In conclusion, COF and wear rate increased with an increase in the printing temperature or layer thickness for ASA-printed specimens. A further polymer being investigated was PEI, which was FFF-printed with different build directions and layer deposition strategies

and tested under dry conditions with an $Fe_3Al_2$ $(SiO_4)_3$ garnet abrasive [91]. The study indicated that printing in the horizontal orientation (X) and a layer deposition strategy of 0° and 90° reduced material wear and concomitantly exhibited a low COF. In [90], PES and PES/BF with BF-fiber-reinforced composite specimens were printed with different printing temperatures via FFF and tribologically characterized. The major outcome of the investigations revealed that the wear rate was reduced by 80%, compared with pure PES, due to fiber reinforcement. At the same time, the width of the wear track was reduced by 45%.

**Table 2.** Overview of research analyzed in tribology of additively manufactured polymers.

| Ref. | Topic | Objective | Application | Manufacturing Process | Methodology | Findings |
|---|---|---|---|---|---|---|
| [72] | Comparison of conventional, milled, and printed PMMA components | Identification of polishability and wear behavior | Biotribology, Occlusal splints | DLP, conventional and subtractive | Wear tests (reciprocating movement; lubricated; steatite) | Similar wear behavior |
| [73] | Tribological behavior of Med610 | Investigating tribological performance | Biomedical | PolyJet | Dry ring-on-disc tribometer with C45 steel counterparts under different loads | Low abrasion resistance |
| [74] | Comparison of different printed orientations of PA12 parts | Identification of wear and frictional behavior | Basic research | SLS | Pin-on-disc (rotatory movement; dry; against 100Cr6-Disc) | Wear increased with contact temperature; 0° print orientation best trib. behavior |
| [75] | Comparison of printing orientations and tribological loadings of TPU and PA12 | Identification of wear and frictional behavior | Shoe soles | SLS | Wear tests (reciprocating movement against steel balls and sandpaper; dry) | COF and wear TPU > PA12; COF raised with sliding speed; Printing orientation no clear difference |
| [77] | Comparison of compression molded and FFF printed ABS | Identification of tribological behavior under different loads | Basic research | FFF | Ball-on-disc (rotatory movement; paraffin oil lubricated; EN31 steel ball) | COF raised with increasing load; COF higher for printed parts; reduced wear rate for higher loads |
| [78] | Comparison of different loads, sliding velocities, and surface configurations for ABS parts | Identification of tribological behavior in contrast to theoretical tribological models | Basic research | FFF | Box-on-plate (reciprocating movement; dry; counterpart ABS) | COF raised with increasing load; reduced with higher velocities; theoretical models provided other findings than experiments |
| [79] | Comparison of printing process parameters of ABS and PLA | Identification of tribological behavior | Basic research | FFF | Block-on-roller (rotatory movement; dry; against EN 8 roller) | ABS parts with higher COF but lower wear rate than PLA parts |

**Table 2.** *Cont.*

| Ref. | Topic | Objective | Application | Manufacturing Process | Methodology | Findings |
|---|---|---|---|---|---|---|
| [80] | Comparison of PLA-, ABS-, and PETG-printed gears regarded to their tribological behavior | Identification of tribological behavior and service life | Gears | FFF | Gear test rig with St 37-2 steel gear (wear/ service life; rotatory movement; dry) | PETG exhibited the best performance and the highest service life |
| [81] | Comparison of printing process parameters for PC-ABS parts | Identification of frictional behavior | Basic research | FFF | Pin-on-disc (rotatory movement; dry; EN 31 steel plates) | Optimal parameters for the lowest COF were found after screening |
| [82] | Comparison of printing process parameters for PC-ABS parts | Identification of wear behavior | Basic research | FFF | Pin-on-disc (rotatory movement; dry; EN 31 steel plates) | Optimal parameters for the lowest wear rate were found after screening |
| [83] | Comparison of printing process parameters of carbon-fiber reinforced ABS | Identification of tribological behavior | Basic research | FFF | Pin-on-disc (rotatory movement; dry and water lubrication; against 40 HM steel plate) | Dry: COF higher with higher fiber infill; Water: COF and wear rate significantly lower with increasing fiber infill |
| [84] | Comparison of silicon filler percentages in PLA composites | Identification of tribological behavior | Basic research | FFF | Pin-on-disc (rotatory movement; dry; counterpart EN19 steel) | Filling with silicon decreased the wear rate and COF |
| [85] | Comparison of printing process parameters of PLA | Identification of tribological behavior | Basic research | FFF | Pin-on-disc (rotatory movement; dry; against $Al_2O_3$ ball) | COF: Black < natural PLA at the same temperature; higher temperature $\rightarrow$ higher COF |
| [86] | Comparison of the tribological behavior of PLA, HT-PLA, and PETG | Correlation between printing parameters and tribological behavior | Basic research | FFF | Cylinder-on-plate (rotatory movement; dry) | HT-PLA: lowest wear rate and COF; vertical printing orientation favored for all parts |
| [87] | Effect of printing orientation and bronze existence on tribological behavior of bronze/PLA composite parts | Evaluation of bronze in PLA composites and the influence on the tribological behavior | Basic research | FFF | Cylinder-on-plate (rotatory movement; dry; against steel plate) | Bronze reduced wear rate but not COF, and vertical printing orientation exhibited the highest COF, but the lowest wear rate; |

| Ref. | Topic | Objective | Application | Manufacturing Process | Methodology | Findings |
|---|---|---|---|---|---|---|
| [88] | Comparison of different printing process parameters of PETG | Identification of tribological behavior | Basic research | FFF | Pin-on-disc (rotatory movement; dry; 100Cr6 steel counterpart) | Higher load and/or vertically printed → higher wear rate; Higher layer thickness → higher COF |
| [89] | Comparison of different printing process parameters for PBT | Identification of tribological behavior | Basic research | FFF | Pin-on-disc (reciprocating movement, dry) | COF was hardly dependent on the degree of infill; low infill → higher wear |
| [90] | Comparison of printing temperature, BF fiber reinforcement of PES | Identification of tribological behavior | Basic research | FFF | Wear tests (rotatory movement; dry) | BF fiber reinforcement improved the wear resistance |
| [91] | Comparison of printing process parameters of PEI | Identification of tribological behavior | Basic research | FFF | Pin-on-disc (rotatory movement; dry; against steel disc) | Low wear rate and COF → horizontal orientation (X) and a layer deposition strategy of 0° and 90° |
| [92] | Comparison of printing temperature and layer thickness of ASA | Identification of tribological behavior | Basic research | FFF | Pin-on-disc (rotatory movement; dry; AISI 304 counterpart) | Higher layer thickness, temperature → higher wear rate and COF |

### 4.2. Metals

Although plastics still have the largest share of the AM market [76], the interest in tribological aspects seems to be significantly higher for metallic materials due to the increase in the number of publications regarding these materials. The most commonly used processes can be roughly divided into powder bed processes and direct extrusion processes according to the type of material feed. The designation of different processes is partly defined differently. The following paragraph therefore briefly summarizes the most important designations.

The most important subdivision is based on the material supply. In powder-bed-based processes, the metallic material is applied in a powder form from a reservoir after each layer. A subsequent energy input heats the loose powder in the second step until the powder melts or is sintered. The fine subdivision of the processes results from the energy input used. The terms **direct metal laser sintering** (DMLS), **selective laser sintering** (SLS), **selective laser melting** (SLM), and **laser powder bed fusion** (LPBF) describe the use of a laser for melting or sintering. When an electron beam is used, the process is referred to as either **electron beam powder bed fusion** (EBPBF) or **electron beam melting** (EBM). All powder-bed-based processes are usually grouped together under the term **powder bed fusion** (PBF). With 58 of the 77 articles examined, most of the submissions fall into this category. In contrast to powder-bed-based processes, there are processes in which the material to be printed is applied directly (**direct energy deposition** (DED)). The subdivision is then further established on the basis of the energy source used and the form of the material

supplied. In **electron beam direct energy deposition** (EBDED), an electron beam melts the material supplied in wire form, creating a bond with the underlying layer. This is in contrast to **laser wire direct energy deposition** (LWDED). Here, the energy input is performed using a laser. If the material is applied into the extrusion area in a powder form, the laser-based process is known as **laser powder bed direct energy deposition** (LPDED) or **laser additive manufacturing** (LAM). Finally, in **wire-arc additive manufacturing** (WAAM), a wire is melted with the help of an electric arc. This is similar to a typical welding process.

Apart from the processes described, there are also other approaches that work with highly filled filaments in **fused filament fabrication** (FFF), for instance. However, since these processes have not yet been investigated with regard to their tribological aspects, they will not be discussed further here. Figure 4 shows the proportion of manufacturing processes used in the contributions examined.

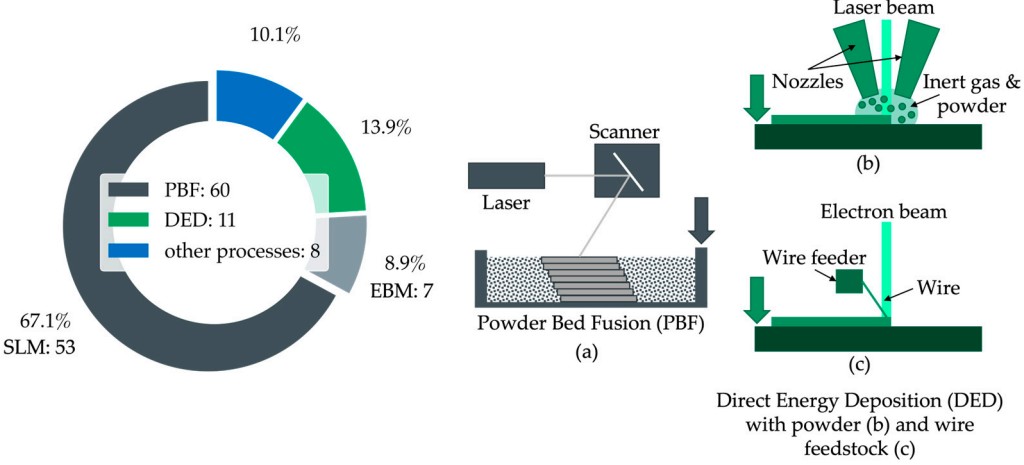

**Figure 4.** Classification in subcategories for metals: (**a**) powder bed fusion (PBF); (**b**) direct energy deposition (DED) with powder; (**c**) wire feedstock.

The following passages summarize previous research in the area of additively manufactured metal materials. The subdivision is based on the different processes. The publications considered are summarized in Table 3.

**Table 3.** Overview of research analyzed in tribology of additively manufactured metals.

| Ref. | Topic | Objective | Application | Manufacturing Process | Methodology | Findings |
|---|---|---|---|---|---|---|
| [93] | Additive manufactured aluminum-based metal matrix composites | Comparison of different composites with samples made by casting | Basic research | DMLS | Pin-on-disc, dry, WC cemented with Co pin | Composites had a lower COF but higher wear with micro size $TiB_2$ reinforcement but higher COF in comparison to samples made by casting |
| [94] | Tribological behavior of 316 L produced using SLM | Compare friction and wear with traditional manufacturing using different counter bodies | Hydraulic application | SLM | Ring-on-disc, L-HM 46 hydraulic oil lubricated, counterpart: H68 brass, ion-nitriding 38CrMoAl, L-HM 46 hydraulic oil | Reduced friction and wear using brass as the counterpart |

**Table 3.** *Cont.*

| Ref. | Topic | Objective | Application | Manufacturing Process | Methodology | Findings |
|------|-------|-----------|-------------|----------------------|-------------|----------|
| [95] | Tribological performance of 316 L | Friction and wear under different build-up directions and test conditions | Basic research | SLM | Linear tribometer, dry reciprocating sliding with E-52,100 hardened steel balls | No significant effect of build-up direction on friction and wear, decreased COF at higher temperatures |
| [96] | Influence of pores and density on tribological performance | Friction and wear in dependence on the process-related density under different contact conditions | Hydraulic application | SLM | Pin-on-disc, mixed and hydrodynamic lubrication, hardened 38CrMoAl Disc, 6 mm diameter 316 L pin, both polished, L-HM46 hydraulic oil | Increased laser exposure time lead to reduced pores and higher hardness, COF decreases with lower density |
| [97] | Additive manufactured tilting-pad journal bearings for improved cooling | Cooling pads with internal channels | Tilting-pad journal bearings | SLS | Test rig with ISO VG46 oil-cooled pads, different cooling conditions, and comparison to TEHL simulations | Multichannel design performed best in terms of temperature reduction |
| [98] | Comparison of manufacturing techniques of M3:2 high-speed steel | Resulting properties of cast steel, hot-isostatic pressing and SLM | Basic research | SLM | Microstructure, hardness, and pin-on-paper wear test against a bonded abrasive paper of cast steel, hot-isostatic pressing and SLM | Tribological behavior is promising compared with conventional manufacturing |
| [99] | Tribological behavior of 17-4 PH | Investigating wear resistance and friction of 17-4 PH | Basic research | LB-PBF | Dry and lubricated tests on ball-on-disc tribometer under 10 N and 30 N with 52,100 steel balls | LB-PBF have higher wear in lubricated but lower wear in dry conditions compared with CM |
| [100] | Plasma oxidation of LPBF-manufactured titanium | Improving mechanical properties and tribological behavior by plasma oxidation | Basic research/ biomedical | LPBF | Microstructure, hardness, pin-on-disc under dry and SFB lubrication with $Al_2O_3$ ball | Better wear resistance compared with forged samples, due to higher hardness, larger diffusion zone, and microstructure |

**Table 3.** *Cont.*

| Ref. | Topic | Objective | Application | Manufacturing Process | Methodology | Findings |
|------|-------|-----------|-------------|----------------------|-------------|----------|
| [101] | Spherical powders with silver content | Improving friction and wear | Basic research | LAM | Ball-on-disc tribometer with $Si_3N_4$ balls under different loads and temperatures | Powders produced via combustion reaction showed better performance compared with the gas atomization process, and the lubrication layer with silver led to low friction and reduced wear |
| [102] | Properties of SLM Al-12Si | Influence of scanning strategy and heat treatment on tribological behavior | Basic research | SLM | Single melt and checkerboard scanning, microstructure, hardness, dry pin-on-disc wear tests with hard-faced stainless steel disc | Untreated SLM samples show lower wear compared to heat-treated and cast samples, higher wear at SM Samples compared to CB due to higher porosity |
| [103] | Micro-arc oxidation (MAO) treatment of SLM Ti6Al4V | Improving wear resistance through MAO | Basic research | SLM | Ball-on-disc dry sliding wear test with WC-CO ball, microhardness | Coating growth slowed down with increasing treatment duration, tribological performance increased through MAO treatment |
| [104] | Characterization of Nickel Aluminum Bronze | Provide high density materials | Basic research | LPBF | Hardness, tensile strength, and reciprocating sliding with JIS SUJ2 steel counterpart and engine oil FG-5 | Higher wear resistance, tensile strength, and hardness compared to conventional manufacturing |
| [105] | Wear resistance of Ti6Al4V for biomedical application | Comparing wear of SLM samples with wrought and heat-treated samples | Biomedical | SLM | Reciprocation ball-on-plate tribometer with $Al_2O_3$ balls and artificial saliva | Higher hardness and wear resistance compared to wrought and wrought heat-treated samples due to microstructure but in the the direction of the molten lines, the wear resistance was the worst |

**Table 3.** *Cont.*

| Ref. | Topic | Objective | Application | Manufacturing Process | Methodology | Findings |
|---|---|---|---|---|---|---|
| [106] | Machinability of additive manufactured Ti6Al4V | Examining the influence of anisotropy for tool wear | Machinability of AM parts | LPBF | Analyzing tool wear and machined surface quality after milling | Tool life decreased by up to 40% depending for vertical manufactured samples |
| [107] | Metal–diamond composites | Increasing wear impact resistance | Mining application | SLM-SPS | Impact abrasive experiments | Impact-abrasive resistance increased with higher Mo-Cr, Ni, and coated diamond content |
| [108] | Increasing hardness via electron beam irradiation and thermal oxidation | Increasing wear resistance of Ti6Al4V | Basic research | SLM | Microstructure, hardness and wear pin-on-disc test under dry conditions with WC ball | Electron beam treatment increased surface roughness and surface layer hardness. Thermal air oxidation resulted in oxide layer buildup. |
| [109] | Heat treatment of 316 L | Investigation influence on microstructure, mechanical behavior and wear | Basic research | SLM | Microhardness, XRD, wear on reciprocating tribometer under dry lubrication with tungsten carbide ball | The dominant influence of porosity on wear behavior, wrought samples showed better wear resistance |
| [110] | Finish-Milling effect on surface and wear | Improving surface, mechanical properties, and wear | Basic research | SLM | Roughness, microhardness, and wear on dry reciprocating tribometer with tungsten carbide ball | Wear rate reduced using finish-milling operation with a high feed rate |
| [111] | Scratch and wear resistance of 316 L | Understanding friction and wear mechanisms of AM 316 L | Basic research | LPBF | Scratch tests under different load, dry sliding ball-on-disc tests with bearing ball | Lower friction and wear compared to conventionally manufactured specimen |
| [112] | Wear of AlCrFeCoNi Alloy coating | Investigating wear behavior | Basic research | SLM | Dry friction and wear test under different loads and speeds with SiC ball | Coating led to excellent hardness and wear resistance |

**Table 3.** *Cont.*

| Ref. | Topic | Objective | Application | Manufacturing Process | Methodology | Findings |
|------|-------|-----------|-------------|-----------------------|-------------|----------|
| [113] | Tribological and tribocorrosion behavior of Co-Cr-Mo alloy | Investigation of temperature influence in post-treatment with ultrasonic nanocrystal surface modification (UNSM) | Biomedical | SLM | Ball-on-disc test with bearing steel ball under dry and NaCl solution lubrication | Better tribological and tribocorrosion behavior at high-temperature UNSM and higher hardness |
| [114] | Tribological behavior of Ti6Al4V coated 316 L SLM with plasma oxidation | Reducing wear | Basic research, Biomedical | SLM | Microhardness, SEM, 3D profilometer, Energy dispersive X-ray spectrometer, and X-ray diffractometer. Wear tests against $Al_2O_3$ balls under dry sliding on a pin-on-disc tribometer | Wear resistance and hardness improved compared to the uncoated 316 L due to the titanium oxide diffusion zone |
| [115] | Corrosion of ceramic-based $TiO_2$ coatings on layered SLM | Reducing corrosion by oxidized Ti6Al4 coating | Biomedical | SLM | Corrosion with open-circuit potential, potentiodynamic polarization, and Electrochemical Impedance Spectroscopy in Simulated Body Fluid, microhardness, SEM, XRD | Improved corrosion resistance compared to untreated 316 L and layered Ti6Al4/316 L, best results with increased oxidation temperature and time |
| [116] | Boride-reinforced steel coating produced by SLM | Characterization of microstructure and wear resistance | Basic research | SLM | Microstructure, hardness, and wear tests | Wear could be reduced significantly by adding the SLM coating, but friction remained similar |
| [117] | Postprocessing of AISI H13 Steel SLM | Investigating the influence of different surface conditions on the tribological behavior | Basic research | LPBF/SLM | Pin-on-disc tribometer against bearing steel with a mineral base oil lubrication | Grinding and polishing lead to the lowest COF, no correlation between COF and surface roughness |
| [118] | Properties of Inconel® 718 produced by L-PBF | Characterization of tribological behavior compared to cast samples | Basic research | LPBF | Microstructure, nanoindentation, and pin-on-disc tribometer under dry lubrication with oil-hardened nickel steel as counterpart | Better mechanical behavior and lower wear compared to cast samples |

**Table 3.** *Cont.*

| Ref. | Topic | Objective | Application | Manufacturing Process | Methodology | Findings |
|---|---|---|---|---|---|---|
| [119] | Heat treatment of Inconel® 718 produced by SLM | Influence of heat treatment temperature on structure, hardness, and wear | Basic research | SLM | Microhardness, Microstructure and wear on reciprocating tribometer under dry conditions with a tungsten carbide ball | SLM-manufactured samples could have higher wear resistance compared to the wrought reference when heat treatment is appropriate |
| [120] | LAM manufactured M50 with Sn-Ag-Cu and $Ti_3C_2$ | Improving tribological behavior and reduce noise | Basic research | LAM | Reciprocating dry sliding tests with $Si_3N_4$ ball at different loads and speeds, noise measurement | M50-Sn-Ag-Cu-$Ti_3C_2$ lead to significant improvement of tribological behavior and reduced noise |
| [121] | Tribological behavior and noise of TC4 with SnAgCu | Influence of SnAgCu concentration on tribological behavior and noise | Basic research | LAM | Reciprocating dry sliding tests with $Si_3N_4$ ball at different loads, noise measurement | 10 wt.% of SnAgCu lead to the lowest COF and noise emission |
| [122] | Process parameters of Ti6Al4V SLM | Investigating the most important printing parameters for the tribological behavior | Basic research | SLM | Pin-on-disc tribometer under dry lubrication with alumina ball | The scanning angle was the most important parameter for the tribological behavior |
| [123] | Printing textured surfaces in SLM | Investigating the influence of surface textures and dimples on the tribological behavior | Basic research | SLM | Ball-on-disc tribometer under dry and lubricant-impregnated conditions with a bearing steel ball | Ball-on-disc tribometer under dry and lubricant-impregnated conditions with a bearing steel ball |
| [124] | TiB/Ti6Al4V composites in LPBF | Improving hardness and wear behavior with boron composite | Basic research | LPBF | Microstructure, microhardness, wear on pin-on-disc tribometer with 60 HRC steel plate | Wear decreased significantly with rising boron content while microhardness improved |
| [125] | High temperature friction and wear of maraging tool steel | Investigating the tribological behavior of maraging steel at high temperatures | Tooling | SLM | Hot-strip drawing tribometer with Al-Si coated 22MnB5 counter surface at 600 °C and 700 °C | Similar friction and wear at 600 °C compared to hot-work steel, unstable friction of maraging steel at 700 °C but similar wear |

**Table 3.** *Cont.*

| Ref. | Topic | Objective | Application | Manufacturing Process | Methodology | Findings |
|---|---|---|---|---|---|---|
| [126] | Effect of shot penning on DMLS 17-4 PH | Influence of shot penning pressure and blasting media on the tribological and corrosion behavior | Basic research | DMLS | Hardness, wear on ball-on-disc under dry lubrication with tungsten carbide-cobalt balls | Steel and ceramic shots improved the hardness and wear resistance most, ceramic shots with 0.6 MPa led to optimum surface morphology, hardness, and microstructure |
| [127] | Graded Ag-multilayer graphene/ TC4 alloy | Friction and wear reduction | Basic research | LAM | Dry sliding tribometer with $Si_3N_4$ ball at different loads and temperatures | Improved friction and wear of graded Ag-multilayer graphene/TC4 alloy compared to homogeneous Ag-multilayer graphene/TC4 alloy and pure TC4 |
| [128] | Friction in additively manufactured fluid channels | Predicting friction factors in SLM manufactured fluid channels | Hydraulic application | LPBF/SLM | Measuring friction factors, Simulation | Friction factors were higher than in classical turbulent flow theory and were influenced by build angle and channel diameter |
| [129] | Anisotropy of Co28Cr6Mo in biomedical application | Investigating the influence of build orientation on mechanical and functional properties | Biomedical | SLM | Tensile testing, microstructure, Fretting tribocorrosion with phosphate-buffered saline, Cytocompatibility | Anisotropy in gain size and morphology led to significant differences in tensile properties and bio-tribocorrosion wear rates |
| [130] | Air-cooling and aging of Ti6Al4V | Influence of post-heat treatment on the tribological behavior | Basic research/ biomedical | LPBF | Microstructure, hardness, and wear on ball-on-disc tribometer with sodium chloride lubricant. | Hardness and wear were improved compared to the conventional manufactured samples |
| [131] | Friction and wear of 316 L under cryogenic conditions | Effect of minimum quantity (MQL) and cryogenic lubrication on the tribological behavior | Basic research | SLM | Ball-on-flat tribometer with 100Cr6 ball under dry, minimum quantity lubrication (MQL), cryogenic, and hybrid cryo + MQL conditions | Wear could be reduced significantly by using a combined cryo + MQL lubrication compared to dry lubrication |

**Table 3.** *Cont.*

| Ref. | Topic | Objective | Application | Manufacturing Process | Methodology | Findings |
|---|---|---|---|---|---|---|
| [132] | Printing parameters in LPBF | Influence of laser power, scanning speed, and hatch spacing on the tribological behavior and thermal stresses | Basic research | LPBF | Direct Impact Hopkinson Pressure bar, dry sliding reciprocating wear test with steel ball | Laser power correlated with thermal stresses, tribological performance improved with higher laser power and lower hatch spacing |
| [133] | TiN coating of ALM Ti6Al4V | Bio-tribology and corrosion resistance compared to wrought reference | Biomedical | ALM | Microhardness, electrochemical measurements, reciprocating sliding wear test | A high-quality and wear-resistant coating could be achieved, corrosion was reduced for the ALM sample compared with the reference |
| [134] | Ferritic-induced high-alloyed stainless steel from LPBF | Investigating wear resistance, microstructure and corrosion | Basic research | LPBF | Hardness, electrochemical potent kinetic reactivation tests, and linearly reciprocating pin-on-plate sliding wear tests | Better wear resistance and hardness and similar corrosion resistance compared to the hot-rolled duplex stainless steel |
| [135] | TC4-Sn-Ag-Cu-Nb2C self-lubricating material with microporous channels | Reducing friction and noise emission | Basic research | LAM | Dry reciprocating tribometer test with $Si_3N_4$ ball, acoustic measurement | TC4-Sn-Ag-Cu-Nb2C in combination with microporous channels improved friction and wear behavior compared to TC4Sn-Ag-Cu and pure TC4, noise emission was also improved |
| [136] | Drag Finish Post-processing of Ti6Al4V | Influence of drag finish processing parameters on the tribological behavior | Basic research | LPBF | Sliding wear test on reciprocating tribometer with $Al_2O_3$ ball and under dry lubrication, hardness, surface roughness | Surface roughness could be reduced significantly compared to the as build state while hardness improved, and wear decreased |
| [137] | Optimizing SLM parameters for 60NiTi alloy | Improving mechanical properties and tribological behavior | Basic research | SLM | Reciprocating sliding tests with $ZrO_2$ balls under dry lubrication | Improvement in wear resistance compared to conventional casting |

**Table 3.** *Cont.*

| Ref. | Topic | Objective | Application | Manufacturing Process | Methodology | Findings |
|------|-------|-----------|-------------|-----------------------|-------------|----------|
| [138] | Wear and tribocorrosion of T6Al4V | Comparing wear and tribocorrosion of SLM and forged Ti6Al4V | Biomedical | SLM | Hardness, microstructure, wear under simulated body fluid, tribocorrosion | SLM samples had better wear resistance than forged ones but a lower corrosion resistance |
| [139] | AM spare parts for hydraulic application | Testing an AM replacement for a slipper-retainer in an axial piston pump | Hydraulic application | SLM | Replacement of the original part and analyzing wear in the application | Successful replacement of the original part without damage to other parts but increase in the surface roughness of other pump parts |
| [140] | Lubrication regimes in AM 316 L | Understanding the influence of the lubrication regime for AM parts | Basic research | LPBF | Ball-on-flat tribometer with 100Cr6 ball, dry, minimum quantity lubrication (MQL), cryogenic and hybrid cryo + MQL conditions | The combination of MQL and cryogenic lubrication helped to provide a good lubrication film and thus improved the tribological behavior |
| [141] | Heat-treatment effect on wear and microstructure | Effect of heat treatment parameters | Basic research | SLM | SEM, EDX, XRD, microhardness and dry sliding wear tests with $Al_2O_3$ balls | Hardness war improves compared with the as-build state and the wear behavior of intercritical heat treatment seemed promising |
| [142] | Protective glass former coating produced by LPBF | Producing wear resistant coatings using LPBF for steel substrate | Coating | LPBF | Microstructure, hardness, and wear tests | It was possible to apply dense and hard coatings with high wear resistance on the substrate while the dominant wear mechanism was abrasive wear |
| [143] | Plasma Electrolytic Oxidation on LPBF Ti6Al4V | Wear resistance in dependency of coating parameters | Biomedical | LPBF | Dry ball-on-plate tribometer with yttria stabilized zirconia counterpart | Increased wear resistance compared to the untreated reference |

**Table 3.** *Cont.*

| Ref. | Topic | Objective | Application | Manufacturing Process | Methodology | Findings |
|------|-------|-----------|-------------|----------------------|-------------|----------|
| [144] | Post-treatment for improving the tribological behavior of lubricated Al-Si alloy | Effect of heat treatment and Electric Discharge Alloying (EAD) | Basic research | SLM | Wear tests on pin-on-disc tribometer with EN-31 alloyed steel counterpart under lubricated conditions at different temperatures | Electric Discharge Alloying (EAD) lead to a significant improvement in wear resistance for SLM and cast specimen, heat treatment improved wear resistance for cast specimen while decreasing for SLM |
| [145] | Carbon nanotube reinforcement of 316 L | Evaluating microstructure and wear | Basic research | LPBF | Microstructure, hardness, and wear in dry sliding tests under different normal loads | Hardness and wear resistance were improved significantly by adding carbon nanotubes |
| [146] | Atmosphere gas carburizing of pure titanium | Improving tribological behavior | Basic research | EBM | Microstructure, hardness, dry friction on pin-on-disc tribometer with a bearing steel ball | Increased hardness and thus reduced wear and COF compared to untreated samples |
| [147] | Tribological behavior of carbide-rich tool steel | Comparison of different carbide contents | Basic research | EBM | Reciprocating ball-on-plate dry wear tests with bearing steel ball, scratch tests, hardness | The lowest COF and wear were observed for the 20% carbide steel compared to the 8% and 25% ones |
| [148] | Wear of ferrite-pearlite steel from EBM | Comparing wear of EBM samples with hot rolled ones | Basic research | EBM | Dry sliding wear test with 52,100 steel balls | Wear resistance of the EBM samples decreased significantly compared to the hot rolled samples due to the changed microstructure |
| [149] | Tribological behavior of EBPBF Ti6Al4V under high temperatures | Evaluation of wear at different temperatures | Basic research | EBPBF | Dry sliding wear against steel and alumina counter bodies | Wear rates decreased with rising temperature |
| [150] | Heat treatment of Ti6Al4V produced by EBM | Effect of different heat treatments on the tribological behavior | Basic research | EBM | Dry sliding wear tests, microhardness | Water-quenching led to the best tribological performance in terms of wear while wear in furnace-cooled samples was higher than in the untreated samples |

**Table 3.** *Cont.*

| Ref. | Topic | Objective | Application | Manufacturing Process | Methodology | Findings |
|---|---|---|---|---|---|---|
| [151] | Fe-Cr-V alloy for wear application | Investigating the possible use for tooling | Basic research/tooling | PBF-EB | Jet-wear test, impact test, field test with AM tools | Less wear in the Jet-wear tests compared to the reference material |
| [152] | Functionally graded material with LAAM | Inconel® 625/SS420 FGM coating of cast steel | Basic research | LAAD | Dry and lubricated reciprocating ball-on-flat tribometer tests with SS440C steel ball, microstructure, hardness | Gradient microhardness and good wear resistance were achieved by the crack-free coating |
| [153] | Wear of Fe-8Cr-3V-2Mo-2W DED | Influence of heat treatment and counterpart | Basic research | DED | Dry Ball-on-disc tests with high-carbon steel and zirconia balls | Wear rates depended on sliding speed and load and lower wear rates for the zirconia counterpart, better wear resistance without heat treatment |
| [154] | Wear and microstructure of Inconel® 718 | Influence of laser power | Basic research | LAM | Microhardness, reciprocating dry wear test with bearing steel ball | Micropores decreased with rising laser power, microhardness, and wear were best for a laser power of 1200 W |
| [155] | Wear of DED M2 tool steel | Wear resistance of M2 compared to conventionally manufactured steel | Basic research | DED | Dry ball-on-disc wear tests with bearing steel and zirconia balls with different speeds and loads | Excellent wear resistance compared to conventionally manufactured steel |
| [156] | Isotropy of mechanical properties and wear behavior | Investigating WAAM-manufactured isotropy | Basic research | WAAM | Tensile testing, dry pin-on-disc wear tests with bearing steel pin | Higher anisotropy was observed with inter-layer cold working in tensile testing and the inter-layer cold working led to a higher hardness and thus improved wear resistance and lower friction |

**Table 3.** *Cont.*

| Ref. | Topic | Objective | Application | Manufacturing Process | Methodology | Findings |
|------|-------|-----------|-------------|----------------------|-------------|----------|
| [157] | Tantalum-zirconium coating on Ti6Al4V | Corrosion and wear resistance | Basic research | DED | Electrochemical measurement in 0.5 mol/L $H_2SO_4$, hardness, dry block-ring wear test with GCr15 ring | Corrosion resistance and surface hardness were improved by the coating and the wear resistance improved significantly |
| [158] | WAAM of Ni-based superalloy | Investigating wear resistance | Basic research | WAAM | Dry sliding wear tests with bearing steel ball at different loads, hardness | Increased hardness and wear resistance of WAAM specimen compared to the wrought ones |
| [159] | DED coating of Fe-12Mn-5Cr-1Ni-0.4C on cast iron | Improving wear resistance | Basic research | DED | Dry ball-on-disc tribometer with bearing steel ball at different loads and speeds, mechanical testing | Wear rates depended mainly on the applied load and at high loads, the manganese steel coating increased wear resistance compared to the cast iron |
| [160] | CoCrFeMoNiV produced by WAAM | Characterizing wear resistance and process parameters | Basic research | WAAM | Mechanical testing, hardness, Miller wear test according to ASTM G75 | It was possible to produce CoCrFeMoNiV with WAAM with good yield strength and wear resistance |
| [161] | Fe-based stainless-steel coating with DED | Corrosion and wear resistance depending on manufacturing speed | Basic research | DED | Microstructure, hardness, corrosion testing, wear test | Corrosion resistance improved with rising manufacturing speed as well as the hardness, wear resistance reaches a threshold at a mean speed level |
| [162] | Lanthanum Oxide in LDD-manufactured Iron-Chromium Alloy | Influence of Lanthanum Oxide on the tribological behavior | Basic research | LDD | Microstructure, dry sliding friction test with silicon nitride ceramic ball | Shorter running in and reduced wear rate with rising $La_2O_3$ content |
| [163] | Hybrid AM with laser cladding | Investigating high-temperature wear | Basic research | Hybrid additive manufacturing | Microstructure, high-temperature tribometer with $Si_3N_4$ ball | Improved wear resistance compared to the substrate |

**Table 3.** *Cont.*

| Ref. | Topic | Objective | Application | Manufacturing Process | Methodology | Findings |
|---|---|---|---|---|---|---|
| [164] | Pre-positioned wire-based electron beam additive manufacturing | Tribological behavior under different loads and main influencing factors | Basic research | Pre-positioned wire-based EBM | Dry pin-on-disc tribometer test with D3 steel counterpart | Load was the dominant factor for the tribological behavior |
| [165] | Cold Gas Spray Additive Manufacturing | Improvement in corrosion and wear resistance | Basic research | Cold Gas Spray Additive Manufacturing | Rubber wheel testing, dry ball on disc tribometer tests with WC-Co ball, corrosion testing | A dense maraging steel part was achieved and a subsequent cermet coating led to a significantly improved sliding wear and water erosion resistance |
| [166] | Inconel® 718 coating by electro-spark deposition | High-temperature wear and corrosion resistance | Basic research | Electro-spark deposition | High-temperature wear and electrochemical corrosion testing | Electro-spark deposition and subsequent boronizing of the H13 tool steel led to improved hardness, wear resistance and corrosion resistance but corrosion resistance was best for the as-deposited coating |
| [167] | Composite coating with Ultrasonic-Assisted Laser Additive Manufacturing | Tribological behavior of TiC composite coating | Basic research | Ultrasonic-assisted LAM | Microhardness, Dry pin-disc friction, and wear tests with hardened 45 steel disc | The content of unmelted TiC was significantly reduced by the ultrasonic assistance and hardness improved compared to the substrate as well as the coating without ultrasonic assistance, but the COF of the coating increased |
| [168] | Comparison of different AM methods | Comparing mechanical properties and tribological behavior | Biomedical | SLM, LENS, WAAM | Tensile testing, pin-on-disc with hardened steel counterpart, hardness | SLM specimen had the best performance in yield and ultimate tensile strength as well as wear resistance while WAAM showed the most ductile behavior |

**Table 3.** *Cont.*

| Ref. | Topic | Objective | Application | Manufacturing Process | Methodology | Findings |
|---|---|---|---|---|---|---|
| [169] | High-temperature wear of AM Ti6Al4V | Temperature influence on tribological behavior | Basic research | SLM, EBM | Linear reciprocating sliding wear tests with WC-CO counterpart at different loads and temperatures | No significant influence of temperature or manufacturing on wear rate |
| [170] | Self-lubricating Al-WS$_2$ composites | Tribological behavior | Basic research | LPBF, spark plasma sintering | Hardness, dry ball-on-flat tribometer with Si$_3$N$_4$ ball | Self-lubricating LPBF showed slightly better tribological behavior compared to SLM-SPS |

### 4.2.1. Selective Laser Melting (SLM)

Among the works examined in this review, powder bed processes are the most used. With a total of 51 contributions, the most common method of heat input is using a laser beam (Figure 4a).

One of the first works in the context of tribology with AM is the work of Lorusso et al. [93] from 2016. The authors compared the tribological behavior of the samples fabricated using DLMS from AlSi10Mg reinforced with TiB$_2$ micro- and nanoparticles. In addition to microstructure and hardness, they investigated tribological behavior on a pin-on-disc tribometer under dry conditions. The microhardness was higher for all the printed samples than for the casted reference. A WC/Co pin with a 3 mm radius was used as the counterpart. The COF was higher for the printed samples than for the casted ones, but the friction was reduced by adding particles. Wear was lowest for the printed samples with nanoparticles but highest when microparticles were used. Another early contribution was that of Zhu et al. [94]. They studied the differences in the tribological behavior of 316 L stainless steel produced using the SLM process depending on the counterpart used and in comparison, with a conventionally produced reference. With a soft brass counterpart, a slight reduction in friction and wear occurred in SLM-printed samples in lubricated tests on a ring-on-disc tribometer. The authors attributed the effect to the grain refinement in the SLM process. With a hardened counterpart made of an ion nitride 38CrMoAl steel, the differences in friction and wear increased, with plastic deformation occurring in the 316 L samples. Another paper dealing with SLM-printed 316 L specimens is from Li et al. [95]. In tests using a linear reciprocating tribometer under dry sliding with E-52100 hardened steel balls, different loads and print orientations were investigated. The print orientation had no significant effect on friction and wear. At higher temperatures up to 600 °C, the COF continued to decrease, while wear reached its maximum at 200 °C and then decreased with increasing temperature. This could be attributed to the formation of a wear-resistant oxide layer at higher temperatures. The influence of surface pores from the SLM process on the tribological behavior of 316 L was investigated in [96]. A shorter laser exposure time increased the porosity of the printed pins. These were tested under mixed lubrication conditions as well as hydrodynamic conditions on a pin-on-disc tribometer against hardened 38CrMoAl discs. Lower density reduced hardness but resulted in improved lubrication and reduced friction in the experiments.

The porosity in the contact area is particularly decisive for tribological behavior. In their work, Chatterton et al. [97] investigated the use of SLS-produced 316 L tilting-pad journal bearings with integrated cooling channels to reduce the temperature in the thermo-elastohydrodynamically lubricated (TEHL) contact and compared the experimental results with TEHL simulations. A multichannel design achieved the best cooling effect. In [98],

M3:2 high-speed steel was produced using the SLM process under modified manufacturing parameters and was compared with conventional cast steel and hot-isostatic-pressed samples. In addition to the analysis of the microstructure and hardness, pin-on-paper wear tests were carried out. The samples produced using the SLM process showed comparable behavior to the conventionally produced samples. The wear of 17-4 PH manufactured in LB-PBF under lubrication and dry conditions was investigated by Nezhadfar et al. in [99]. The additively manufactured specimens showed less wear under dry conditions and higher wear under lubricated conditions, compared with the conventionally manufactured specimens. To improve the tribological behavior, Kovaci et al. [100] investigated the effects of plasma oxidation on friction and wear in LPBF-fabricated titanium (Ti) specimens compared with forged ones. In addition to the analysis of the microstructure and hardness, pin-on-disc tribometer tests were performed under dry conditions and with simulated body fluid. Due to the rougher surface, friction was higher in the additively manufactured specimens, but the wear rate was lower due to higher hardness, a deeper $TiO_2$ diffusion layer, and changes in microstructure. Li et al. [101] investigated the influence of a spherical Ti6Al4V powder with 10% silver (Ag) content. The Ag formed a lubrication layer, which led to less friction and reduced wear. The powders produced in the combustion reaction showed improved tribological behavior compared with the powders from the gas atomization process. The influence of the scanning strategy and a subsequent heat treatment in the SLM process was investigated using the example of the material Al-Si$_{12}$ in [102] in comparison with cast samples. The untreated SLM samples showed the lowest wear rate in the pin-on-disc test. The use of the checkerboard scanning strategy resulted in less wear compared with a single melt strategy due to less porosity.

One promising approach to improve the tribological behavior is to reinforce the additively manufactured components with a ceramic or ceramic-based coating or targeted oxide formation. By using micro-arc oxidation, Yan et al. [103] succeeded in significantly reducing the wear of Ti6Al4V produced using the SLM process in dry wear tests on a ball-on-disc tribometer. With longer treatment durations, the wear decreased further. The roughness increased, and the layer buildup decreased with a longer treatment duration. Alkelae et al. [104] examined the potential of the LPBF process to produce nickel–aluminum–bronze with a high density and improved tribological behavior and mechanical properties. The authors compared the results with other values in the literature and reported improved mechanical properties and tribological behavior.

To investigate the potential of AM in biomedical applications, Zhou et al. [105] compared the wear behavior of SLM-produced Ti6Al4V in tribometer tests with forged samples. Artificial saliva was used as a lubricant. In addition to a higher hardness rate of the SLM specimens due to changes in their microstructure, less wear was observed. However, this behavior is strongly direction-dependent. The wear of the SLM specimens was highest when the tests were carried out in the direction of the melted paths. In many cases, the post-treatment of additively manufactured samples is necessary. Therefore, Lizzul et al. [106] studied the same material Ti6Al4V with respect to its machinability. Due to the anisotropy of the material, a vertical printing orientation led to a significantly increased wear on the milling tool compared with a horizontal printing orientation. An influence on the surface quality of the post-processed parts was also evident. For use in mining applications, Rahmani et al. [107] investigated how functional lattices filled with metal diamond composites can be produced in a combined SLM process with spark plasma sintering (SPS). In experiments, it was shown that wear can be increased using a suitable material composition. Another approach for hard, wear-resistant components produced using the SLM process was explored in [108] using Ti6Al4V as an example. Electron beam irradiation significantly increased the hardness of the surface and thermal air oxidation led to very hard oxide layers on the surface but also to an increased surface roughness. By combining both processes, a particularly high wear resistance was achieved. The relationship between the wear behavior and the subsequent heat treatment and the resulting porosity of SLM-printed 316 L specimens was examined in [109]. Wear could be reduced through

heat treatment at higher temperatures. The authors considered the reduced porosity as the main reason. However, in dry-running tribometer tests, the conventionally manufactured specimens had the highest wear resistance. Mechanical post-treatment is also suitable for reducing the wear of AM components. Tascioglu et al. [110] compared the wear of Inconel® 625 samples from SLM depending on the process parameters of a finish-milling post-treatment process. A high feed rate led to the best wear behavior in the dry reciprocating tribometer tests. In their work, Upadhyay et al. [111] compared the tribological behavior of 316 L on a ball-on-disc tribometer additionally with scratch tests, where the determined COFs were slightly lower than the tribometer. The samples produced using the LPBF process showed less friction and wear than conventionally produced samples. Yang et al. [112] used an AlCrFeCoNi high-entropy alloy to coat a stainless steel substrate. The tests demonstrated high hardness and wear resistance of the coating.

In many technical applications, the focus is on investigating the wear behavior. However, in applications where corrosive environmental conditions are present, a separate investigation of corrosive and tribocorrosive behavior is also of crucial importance. This applies in particular to applications in the biomedical field, for example, in implants. In [113], the tribological and tribocorrosive behavior of a Co-Cr-Mo alloy produced using the SLM process was improved via post-treatment with ultrasonic nanocrystal surface modification (UNSM). The behavior was best with post-treatment at a high temperature of about 500 °C. Additionally, for possible biomedical applications, Tekdir et al. [114] investigated a coating of Ti6Al4 applied to 316 L using the SLM process, which was additionally plasma-oxidized to produce a wear-resistant ceramic coating. In dry tribometer tests on $Al_2O_3$ balls, a significantly improved wear resistance and hardness could be observed. In [115], the authors subsequently investigated the corrosion behavior of 316 L coated with Ti6Al4V. After fabrication using the SLM process, the plasma oxidation of the surface was performed to produce a ceramic $TiO_2$ coating. This significantly improved the corrosion behavior of the surface. Hardness and corrosion resistance increased with treatment time and temperature. Freitas et al. [116] used the SLM process to apply only a boride-reinforced thick coating on a low-carbon steel substrate. Wear was significantly reduced by the coating, but friction remained at a similar level. This demonstrates the potential of additive processes also for the post-treatment of conventionally manufactured components.

One challenge of additively manufactured components in tribological applications is their rough surface. Guenther et al. [117] therefore investigated how different post-treatments affect pin-on-disc tribometer tests. The SLM-produced H13 steel samples had the lowest friction rate when they were ground and polished. Polishing alone, as well as subsequent laser texturing, increased friction compared with the initial printed condition. Jeyaprakash et al. [118] investigated Inconel® 718 in their work. The additively manufactured LPBF specimens showed improved mechanical behavior and less wear in the tribometer test under dry lubrication with oil-hardened nickel steel as the counterpart, compared with the cast specimens. The influence of heat treatment with different temperatures on the latter material was studied in [119]. Compared with wrought reference specimens, a lower wear rate was achieved in tribometer tests under dry conditions with a tungsten carbide ball, provided that the heat treatment would be correctly designed. The selection of sufficient printing parameters is also relevant for the tribological performance of SLM components. In [120], in addition to the tribological behavior, the friction-induced noise emission of LAM-manufactured M50 samples was investigated when they were alloyed with Sn-Ag-Cu and $Ti_3C_2$. The best tribological behavior as well as the minimum noise emission could be determined for the M50-Sn-Ag-Cu-$Ti_3C_2$ samples. Similar tests were carried out by Quin et al. [121] for TC4 with different content of SnAgCu. The samples with a content of 10 wt.% of SnAgCu achieved the best results in terms of tribological behavior and noise emission. Sagbas et al. [122] explored how the parameters of scanning angle, laser power, scanning speed, and hatch distance affect the tribological behavior of Ti6Al4V. The scanning angle had the greatest influence on the statistical evaluation, followed by the laser power. In [123], the effect of hexagonal prismatic textures directly applied in the

SLM printing process on the tribological behavior of 17-4 PH was investigated. Friction and wear rates were lowest for the non-textured samples. However, the influences of the texture geometry on the behavior could be observed. The manufacturing of composites in LPBF was investigated by Verma et al. [124]. The addition of boron to Ti6Al4V significantly increased microhardness and wear resistance in tribometer tests with increasing boron content. For application in additively manufactured tools, the high-temperature behavior of SLM-produced maraging steel was investigated in [125] in comparison with conventional hot-work steel. At 600 °C, friction and wear in the hot-strip drawing tribometer test were comparable. At 700 °C, the maraging steel failed earlier due to unstable friction.

One possibility to improve the tribological behavior of DMLS components is subsequent shot peening. In [126], the authors investigated how the wear and corrosion of specimens fabricated of 17-4 PH changed when they were shot-peened at different pressures and with different blasting media. Shot peening with ceramic balls under a pressure of 0.6 MPa led thereby to the best results. The potential of graded Ag-multilayer graphene/TC4 alloy in tribological applications was determined in [127]. Compared with pure TC4, as well as homogeneous Ag-multilayer graphene/TC4 alloy, improved friction and wear behavior at different loads and temperatures could be demonstrated. For hydraulic applications, Zhou et al. [128] investigated the influence of the printing direction and tube diameter on the friction factors of 316 L tubes produced using SLM. The friction factors were higher than those expected from classical turbulent theory. Both the pressure direction and the diameter had an influence on the roughness and thus the flow behavior. In addition to the microstructure and mechanical properties, Acharya et al. [129] investigated the tribocorrosion behavior and cytocompatibility of SLM-produced Co28Cr6Mo for implant applications. The anisotropy in grain size and morphology resulted in a significant difference in mechanical properties, as well as bio-tribocorrosion behavior and cytocompatibility, depending on the printing orientation. The influence of air cooling and aging following heat treatment was investigated in [130] for Ti6Al4V. In addition to a changed microstructure, hardness and wear resistance could be increased compared with conventionally produced samples. The effects on the tribological behavior of cryogenic lubrication were studied by Demirsöz et al. [131] for 316 L steel produced with SLM. Compared with dry lubrication, minimum quantity lubrication (MQL) and cryogenic lubrication achieved a significant reduction in wear. The effect was even more pronounced with combined cryo + MQL lubrication. For the same material, the influence of printing parameters on thermal stresses and tribological behavior was investigated [132]. Thermal stresses correlated most with changes in laser power. Tribological behavior improved at high laser power and low hatch spacing. The possibility of a TiN coating on an additively printed Ti6Al4V substrate was analyzed by Esfahani et al. [133] for biomedical applications. Compared with the wrought reference, a very good wear-resistant coating could be produced, and even the corrosion was reduced. The potential for tribological applications resulting from microstructure transformation was investigated by Freitas et al. [134] in LPBF. Starting from 2205 duplex stainless steel, the powder was printed without heat treatment, resulting in a ferritic microstructure. Hardness and wear resistance were higher than for the hot-rolled duplex stainless steel, and the corrosion resistance was comparable. In their work, Gao et al. [135] prepared a TC4-Sn-Ag-Cu-Nb2C alloy in combination with microporous channels using the LAM process to reduce friction and noise emissions. Compared with the pure TC4 and TC4-Sn-Ag-Cu, friction and wear as well as noise emissions were reduced in tribometer tests. In another work by Güneşsu et al. [136], the authors dealt with the mechanical reworking in drag finish on LPBF samples manufactured using Ti6Al4V. In addition to a significantly improved surface roughness, a higher hardness and thus reduced wear could be achieved. Guo et al. [137] optimized the manufacturing parameters for the material 60NiTi using the SLM process. In reciprocating sliding tests, improved wear resistance was achieved, with suitable parameters compared with the cast samples. Huang et al. [138] performed a systematic comparison of SLM specimens and forged ones fabricated using Ti6Al4V

under simulated body fluid lubrication. While the SLM specimens showed improved wear resistance, the resistance to tribocorrosion was lower.

AM has great potential in the rapid provision of spare parts. For this reason, Klimek et al. [139] investigated the performance of a slipper–retainer in an axial piston pump under application conditions. Further operation with the additively manufactured component was successful. However, the roughness of the other pump parts increased during the test. To achieve an understanding of the prevailing lubrication regime in additively manufactured LPBF components, in [140], the authors investigated how different types of lubrication affect the lubricant film structure and thus the tribological behavior. An elastohydrodynamic lubrication (EHL) regime was achieved with simultaneous cryogenic and minimum quantity lubrication, which is why the tribological behavior was best there. The influence of heat treatment parameters was investigated in [141] for maraging steel produced using SLM. Hardness was improved in all cases, and it was shown that an intercritical heat treatment is promising. In [142], the LPBF process was used to produce wear-resistant glass former coatings on steel substrates. The coatings were able to exhibit high hardness and density, as well as high wear resistance. Santos et al. [143] investigated a possible coating via the plasma electrolytic oxidation of Ti6Al4V for biomedical application and demonstrated improved wear resistance compared with the untreated samples. The post-treatment of Al-Si alloy via heat treatment and electric discharge alloying (EDA) was investigated in [144]. The EDA treatment resulted in reduced wear for both the SLM specimens and the cast specimens, while heat treatment resulted in a positive effect only for the cast specimens. To reduce the wear of 316 L steel produced using LPBF, Yin et al. [145] used carbon nanotube reinforcement. This resulted in increased hardness of the material and, as a consequence, significantly reduced wear.

### 4.2.2. Electron Beam Melting (EBM)

The second powder-based process considered in this review is EBM. In contrast to SLM, the powder is melted with an electron beam. In total, seven articles dealing with this process were examined in this literature review. Among the first papers dealing with the EBM process in a tribological context are those by Kim et al. [146]. The authors investigated the influence of atmosphere gas carburizing on the tribological behavior of pure titanium on a pin-on-disc tribometer. Post-treatment significantly increased the hardness, which reduced the wear compared with the untreated samples. Friction was also less than in the untreated samples and conventional titanium. Iakovakis et al. [147] examined the influence of different carbide contents on the tribological behavior of tool steel in ball-on-plate dry wear tests and scratch tests. The steel with a carbide content of 20% showed the best tribological behavior in terms of friction and wear compared with the 8% and 25% samples. Shamarin et al. [148] compared the wear behavior of 13Mn6 produced using EBM with hot-rolled samples. However, a significantly increased wear resistance of the additively manufactured samples was demonstrated in tribometer tests. The effect of high temperatures on the tribological behavior of Ti6Al4V was studied in [149]. Due to the formation of stable oxide glaze layers, the wear mechanism changed at higher temperatures and, as a result, the wear of the samples was reduced. Subsequent heat treatment of Ti6Al4V samples was studied in [150]. While water-quenching improved the tribological and wear behavior under dry conditions, this was worsened due to cooling in the furnace. For the application in tooling, Franke-Jurisch et al. [151] investigated the wear behavior of Fe-Cr-V alloy in jet-wear and impact tests as well as in field tests with AM tools. In the jet-wear tests, the additively manufactured specimens achieved a higher wear resistance than the reference specimens.

### 4.2.3. Direct Energy Deposition (DED)

While most of the studies to date have dealt with additive processes using powder beds, the number of studies focused on direct processes is comparatively small. Within the scope of this literature study, eleven papers were found that dealt with DED (Figure 4b,c).

Among the first works considering the LAAM process is the one by Liu et al. [152]. The authors investigated the coating of a cast iron with Inconel® 625/SS420 FGM. The Inconel® 625 coating between cast iron and SS420 produced a crack-free coating that showed good wear behavior under both dry and lubricated conditions. In [153], the material Fe-8Cr-3V-2Mo-2W was printed using the DED process and then heat-treated. In tribometer tests with high-carbon steel and zirconia balls, the zirconia counter bodies lead to less wear. The heat treatment resulted in poorer wear behavior regardless of the counterpart used. The influence of laser power on wear behavior was investigated for the Inconel® 718 superalloy using the LAM method [154]. Increased laser power reduced the micropores, and at a power of 1200 W, the highest hardness and the minimum wear rate could be observed. The influence of heat treatment on the wear behavior of M2 tool steel produced using DED was investigated by Park et al. [155]. Compared with conventionally produced samples and high wear-resistant steel produced using DED, M2 showed higher wear resistance. The heat treatment also reduced wear in some load cases. Parvaresh et al. [156] investigated the anisotropy of WAAM-produced stainless steel 347 in terms of mechanical properties and tribological behavior as a function of the manufacturing process. Inter-layer cold working led to increased anisotropy in the tensile tests and at the same time to the higher hardness of the specimens, which resulted in increased wear resistance and lower friction. Xie et al. [157] used the DED process to deposit a wear- and corrosion-resistant tantalum–zirconium layer on a Ti6Sl4V substrate. The generated coating showed higher hardness and corrosion resistance in a 0.5 mol/L $H_2SO_4$ solution and significantly improved wear resistance. For the use of WAAM in the field of Ni-based superalloys, Chigilipalli et al. [158] compared the wear behavior of printed samples with wrought references and were able to demonstrate increased hardness as well as wear resistance in dry sliding wear tests. In [159], the DED process was also used to create a coating on a substrate surface. The substrate used was cast iron, on which a high-manganese steel was deposited. The wear behavior was strongly load-dependent but hardly speed-dependent, and at high loads, the high-manganese steel coating significantly improved the wear behavior.

In contrast to the previously mentioned work, Treutler et al. [160] fabricated samples of CoCrFeMoNiV from wire material using the WAAM process. By suitable process control, it was possible to achieve good yield strength and wear resistance in the manufactured samples. In their work, Xu et al. [161] used the DED process to produce thin Fe-based stainless-steel coatings and investigated the effect of printing speed on corrosion and wear resistance. Higher speeds led to higher hardness and improved corrosion resistance. The wear resistance, however, could not be further improved above a certain level. Zhao et al. [162] investigated the effect of lanthanum oxide on the tribological behavior of additively manufactured LDD iron–chromium alloy samples. A shorter running-in time and reduced wear were observed with increasing $Ta_2O_3$ content.

### 4.2.4. Special and Combined Processes

The previous sections summarized the studies that dealt with the usual direct application processes, as well as powder beds. In addition, there are some contributions with a tribological context that cannot be classified under the typical AM manufacturing processes. These are summarized below.

Hao et al. [163] fabricated an Inconel® 625 layer on an H13 substrate using a hybrid AM process, in which a layer created via multilayer laser cladding was subsequently CNC milled and post-treated with ultrasonic surface rolling. The generated layer showed improved wear behavior in high-temperature tribometer tests. An electron-beam-based method for joining the prepositioned wires of Ti6Al4V was presented by Manjunath et al. [164]. They investigated the influence of load, velocity, and sliding distance on the tribological behavior of the fabricated specimens in an extensive experimental design and created a predictive model. The dominant factor was the load. Vaz et al. [165] compared and combined different thermal spraying processes and materials to produce a wear- and corrosion-resistant surface on steel substrates. A dense maraging layer was produced

via cold-gas spray coating and the deposition of cermets using high-velocity oxyfuel significantly reduced sliding wear and water erosion. Kayalı et al. [166] used electro-spark deposition to produce Inconel® 718 coatings on H13 tool steel. The coatings were subsequently boronized, which further improved hardness and wear resistance. However, the as-deposited coating had the best corrosion resistance. A composite coating with TiC was produced by Niu et al. [167] on a Ti6Al4V substrate using an ultrasonic-assisted LAM process. The use of ultrasound significantly reduced the amount of unmelted TiC and increased the hardness, but the COF also increased.

### 4.2.5. Comparison of Different AM Processes

Most of the work investigated dealt with the tribological behavior of additively manufactured metal components from a special process. In addition, there were three papers in this systematic literature review that compared different manufacturing processes. Using the example of pure titanium for medical applications, Attar et al. [168] compared the SLM, LENS, and WAAM processes in terms of the resulting mechanical and tribological behavior. The samples produced using the SLM process had the highest yield and ultimate tensile strength as well as the highest hardness and, as a result, the best wear resistance. The specimens manufactured using the WAAM process, on the other hand, showed the highest ductility and the lowest wear resistance. In contrast, Li et al. [169] found no significant differences in wear for the titanium alloy Ti6Al4V produced using SLM, EBM, as well as conventionally. They also investigated the behavior at high temperatures and showed that the wear rate was not significantly affected by temperature, regardless of the process. Finally, in [170], self-lubricating Al-WS$_2$ composites were produced using LPBF as well as spark plasma sintering. In dry tribometer tests, the LPBF samples exhibited slightly better tribological behavior.

### 4.3. Ceramics/Cermets

The third group of materials considered in the evaluation of the literature deals with ceramics and their integration into metallic matrices, the so-called cermets. There is no further subdivision, as only 6 of 103 publications are classified in this section. The publications considered in the following are summarized in Table 4. Basically, the research on the tribological behavior of additively manufactured ceramics and cermets deals with aluminum oxide (Al$_2$O$_3$) [171,172], zirconium oxide (ZrO$_2$) [172,173], silicon carbide (SiC) [174], tungsten carbide/cobalt (WC/Co) [175] and chromium carbide (CrC) in a WC/Co matrix [176]. These are produced using different AM processes depending on the raw material as described in previous sections: SLM [171,175], EBM [176], DLP [172,173], SLA [173], binder jetting [172,174], and material jetting [172].

Gu et al. [171] investigated the tribological behavior of Al$_2$O$_3$/AlSi10Mg composite material on a ball-on-disc tribometer under dry lubrication conditions against a GCr15 bearing steel ball at varying loads, sliding speeds, and during a long-term test. The results showed, on the one hand, a higher COF and wear rate with increasing load and, on the other hand, a lower COF and wear rate with increasing sliding speed. In that case, the dominant wear mechanisms identified during the tests related to oxidation wear and abrasive wear. A second study by Schiltz et.al. [172] was also focused on Al$_2$O$_3$ and compared the tribological behavior to ZrO$_2$. The ceramic specimens were printed in three different AM processes (photopolymerization, binder jetting, and material jetting) and were tribologically analyzed on a pin-on-disc tribometer against counterparts, which were flat discs of Al$_2$O$_3$ and ZrO$_2$ under water lubrication. The tribological behavior of ZrO$_2$ revealed no significant difference, independent of the printing process. With respect to Al$_2$O$_3$, photopolymerization was identified as the most favored in terms of tribological performance. Another study by Kim et al. [173] relates not to basic research but to an application of dental prostheses (dental crowns) fabricated using zirconia. The specimens were printed using DLP and SLA processes and were compared with conventional milled ones. Wear tests were performed on a chewing simulator up to 120,000 cycles with a

stainless-steel spherical indenter, and subsequently, the fracture strength was determined. Preloading in the chewing simulator did not result in a significant difference in terms of volume loss. The additively manufactured zirconia crowns performed comparably or better in terms of fracture strength and antagonist wear after simulated masticatory movement compared with the milled crowns. Amanov et al. [174] printed SiC specimens using binder jetting process and afterwards applied a thin layer to the substrate surface using the ultrasonic nanocrystal surface modification technique. Wear tests were performed on a reciprocating motion tribometer against a $Si_3N_4$ ball. Very low wear coefficients ($2.61 \times 10^{-12}$ $mm^3$/Nm) were observed compared with additively manufactured polymer or metal substrates. The applied coating marginally reduced the wear rate at room temperature but significantly reduced it to $6.46 \times 10^{-13}$ $mm^3$/Nm at higher temperatures. The reduction in COF was not significant.

The study by Köhn et al. [175] focused on the material class of cermets. They additively manufactured tungsten carbide/cobalt (WC/Co) specimens using the SLM process and compared the tribological behavior with physical vapor deposition (PVD)-coated (WC/Co) steel substrates. In wear tests (reciprocating sliding) against WC/Co balls, the additively manufactured specimens outperformed the conventional PVD coatings in terms of wear rate. After the run-in phase, a similar coefficient of friction of ~0.35 was observed. Another paper by Iakovakis et al. [176] compared EBM-printed CrC-rich WC-Co carbide specimens under various tribological conditions. Ball-on-disc tests were performed using steel (100Cr6) balls and ceramic ($Al_2O_3$) balls at different load conditions. The use of ceramic balls revealed no load dependence with respect to COF, in contrast to a higher wear rate at higher loads. When a steel ball served as a counterpart, the COF increased with increasing load, but surprisingly, the wear rate was reduced by 56.4%.

**Table 4.** Overview of research analyzed in tribology of additively manufactured materials (ceramics and cermets).

| Ref. | Topic | Objective | Application | Manufacturing Process | Methodology | Findings |
|------|-------|-----------|-------------|----------------------|-------------|----------|
| [171] | Comparison of load, sliding velocity, and long-time test of Al-based ceramics | Identification of tribological behavior and influence of test parameters | Basic research | SLM | Ball-on-disc (rotatory movement; dry; counterpart GCr15 bearing steel ball) | Higher load → higher wear and COF; higher velocity → lower wear and COF |
| [172] | Comparison of printing processes of $Al_2O_3$ and $ZrO_2$ | Identification of the influence of the printing process on tribological behavior | Basic research | Photo-polymerization, Binder Jetting, Material-Jetting | Pin-on-disc (rotatory movement; water lubricated; counterparts flat discs of $Al_2O_3$ and $ZrO_2$) | $ZrO_2$: no significant difference for the printing processes; $Al_2O_3$: Photopoly-merization exhibited the best tribological performance |
| [173] | Comparison of printing methods and milled $ZrO_2$ | Identification of the wear behavior of printed/milled dental prostheses | Biomedical | DLP, SLA | Wear test (chewing simulator; dry; stainless-steel spherical indenter) | Zirconia crowns printable; comparable/better wear behavior than milled specimens |

**Table 4.** *Cont.*

| Ref. | Topic | Objective | Application | Manufacturing Process | Methodology | Findings |
|---|---|---|---|---|---|---|
| [174] | Comparison of USNM coatings on SiC parts | Identification of the influence of the USNM coatings on tribological behavior | Basic research | Binder Jetting | Wear tests (reciprocating movement; lubricated with jet fuel; counterpart $Si_3N_4$ ball) | UNSM coatings reduced wear rate and COF |
| [175] | Comparison of printed WC/Co samples in contrast to PVD coated steel samples | Identification of tribological behavior of the WC/Co printed samples and PVD coatings | Basic research | SLM | Wear test (reciprocating movement; dry; WC/Co ball) | WC/Co-printed samples outperformed PVD coating in wear rate |
| [176] | Comparison of loads and counter bodies of printed cemented carbides parts | Identification of the tribological behavior | Basic research | EBM | Ball-on-disc (reciprocating movement; dry) | $Al_2O_3$ ball: COF no loading dependence, but the wear rate increased with load; 100Cr6 ball: higher COF with higher load, but lower wear rate |

## 5. Conclusions and Outlook

Within the scope of our literature analysis, it was possible to identify some current focal points in which most tribological research is currently taking place in combination with additive manufacturing. On the process side, it can be stated that the EAM process is most frequently used for plastics, accounting for 16 of the 20 articles examined. The reason for this is that this is a comparatively simple and inexpensive process with high market penetration, even in the private user sector. In the metal sector, powder-based processes are used most frequently, with the SLM process accounting for the largest share, with 53 of the 79 contributions considered. Here, it can be assumed that the high interest in the SLM process is related to its frequent use in industry. In the case of ceramic AM processes, on the other hand, no trend toward a particular process can be observed, and with a total of six contributions, this was also the lowest proportion.

Regarding the application focus, it was possible to determine that in all three material categories examined, the focus is still on the scientific fundamentals and the understanding of friction and wear mechanisms of additively manufactured components, even if a fine subdivision was not completely possible here in some cases. In the case of plastics, in addition to 16 contributions from the field of basic research, 2 contributions focusing on biomedical research were also identified. With a total of 11 contributions, there was also a comparatively high level of interest in the area of metallic materials. In addition, there were four contributions with a reference to hydrodynamic applications. Additionally, in the case of ceramic materials, not only one contribution related to biomedical applications but also contributions from the field of basic research could be identified. An overview of the applications investigated can be found in Figure 5. In the future, it will be interesting to observe to what extent the knowledge gained from basic research in the last few years can be transferred to applications. It will also be exciting to see in which tribological applications AM finds a foothold besides biomedical ones.

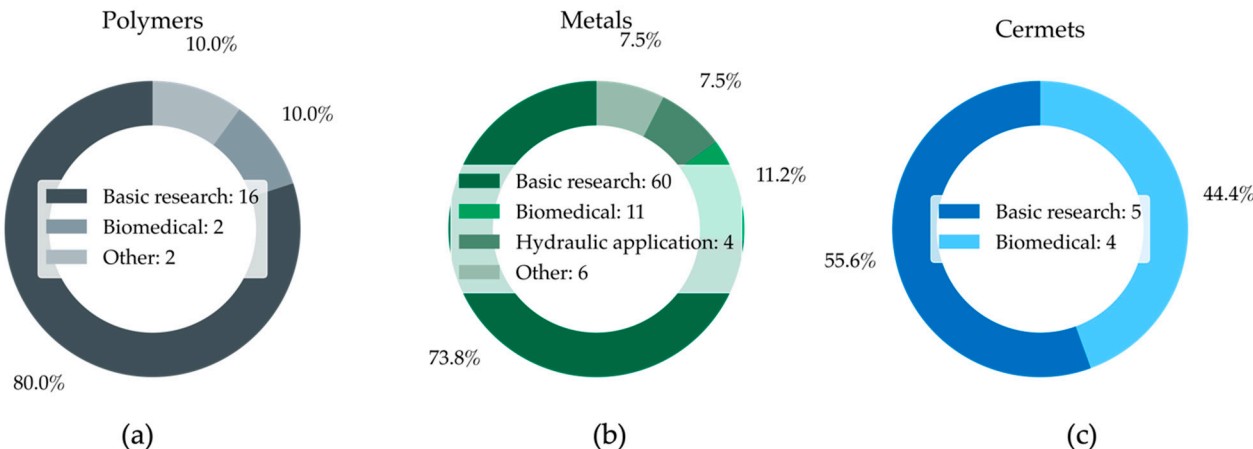

**Figure 5.** Overview of the applications addressed with (**a**) polymers, (**b**) metals, and (**c**) cermet materials.

In addition to applications and objectives, it is also interesting to see which materials are currently used in the tribological context of additive manufacturing. In the case of plastics, ABS (seven contributions), PLA (six contributions), and PETG (three contributions) were studied most frequently. In the case of metallic materials, the focus was primarily on titanium materials, with a total of 25 contributions, with the alloy Ti6Al4V being investigated in 21, followed by stainless steel, with 18 contributions. There, 316 L was used in 14 contributions. In addition, different types of steel (16 contributions) and Ni-based materials (10 contributions) were investigated more frequently. In the case of ceramic materials, no meaningful listing could be established due to the few contributions as well as the mixed materials used. The use of individual materials in AM processes is often also related to which filaments, powders, or other base materials are available for the equipment used. The proportion of materials used is shown in Figure 6 for metal and polymeric materials.

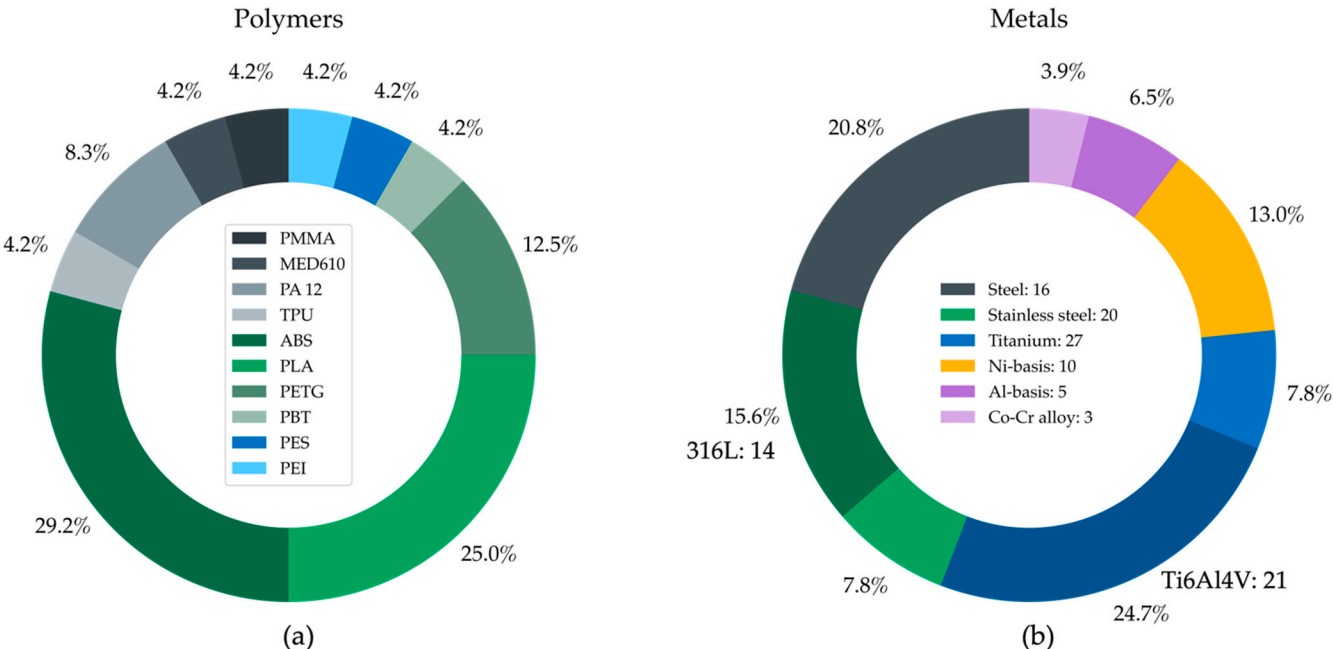

**Figure 6.** Overview of the used materials for (**a**) polymers and (**b**) metals.

Based on this analysis, it is assumed that tribological behavior will increasingly come into focus due to the growing interest in AM processes and their expected use in areas such as sustainable spare part supply, individual products, or biomedical applications.

Furthermore, the following future trends in the field of tribology of additively manufactured components can be expected:

- Further fundamental research to understand how material selection, manufacturing process, and post-treatment affect the tribological behavior of additively manufactured components and the underlying mechanisms;
- The optimization of the manufacturing process, raw material, and post-treatment that can achieve the best possible tribological behavior in the use case, as well as a comparison of different processes in a use case to determine the ideal process
- Applications in the field of individual products, such as in medical technology or the sustainable and rapid provision of spare parts, such as tools or machine elements;
- The generation of wear- and friction-optimized surface layers applied by means of AM processes, as well as the subsequent coating of additively manufactured components using additive or conventional coating processes;
- The functionalization of components in tribological applications through, e.g., integrated cooling or lubrication channels or additional functional integration through the possible design freedom.

**Author Contributions:** Conceptualization, C.O., A.S., T.R. and S.T.; methodology, C.O., A.S., T.R., and S.T.; software, C.O., A.S., and T.R.; formal analysis, C.O., A.S., T.R., and S.T.; investigation, C.O., A.S., and T.R.; resources, S.T.; data curation, C.O., A.S., and T.R.; writing—original draft preparation, C.O., A.S., and T.R.; writing—review and editing, S.T.; visualization, C.O., A.S., and T.R.; supervision, S.T. All authors have read and agreed to the published version of the manuscript.

**Funding:** This research received no external funding.

**Data Availability Statement:** No new data were created or analyzed in this study. Data sharing is not applicable to this article.

**Acknowledgments:** C. Orgeldinger, A. Seynstahl, T. Rosnitschek, and S. Tremmel greatly acknowledge the continuous support of the University of Bayreuth.

**Conflicts of Interest:** The authors declare no conflict of interest.

## Nomenclature

| | |
|---|---|
| ABS | Acrylnitril–butadien–styrol |
| ALM | Additive layer manufacturing |
| AM | Additive manufacturing |
| ASA | Acrylnitril–styrol–acrylester |
| COF | Coefficient of friction |
| DED | Direct energy deposition |
| DLP | Digital light processing |
| DMLS | Direct metal laser sintering |
| EAM | Extrusion-based additive manufacturing |
| EBM | Electron beam melting |
| EB-DED | Electron beam direct energy deposition |
| EB-PBF | Electron beam powder bed fusion |
| EDA | Electric discharge alloying |
| FFF | Fused filament fabrication |
| GMAW | Gas metal arc welding |
| LAAM | Laser-aided additive manufacturing |
| LAM | Laser additive manufacturing |
| LBF | Laser bed fusion |
| LDD | Laser direct deposition |

| LENS | Laser-engineered net shaping |
| LB-PBF | Laser beam powder bed fusion |
| LPBF | Laser powder bed fusion |
| LPBF-M | Laser powder bed fusion–metal |
| LP-DED | Laser powder direct energy deposition |
| MAO | Micro-arc oxidation |
| MQL | Minimum quantity lubrication |
| PA | Polyamide |
| PBF | Powder bed fusion |
| PBT | Polybutylene terephthalate |
| PC | Polycarbonate |
| PEI | Polyetherimide |
| PES | Polyether sulfone |
| PETG | Polyethylene terephthalate glycol |
| PLA | Polylactic acid |
| PMMA | Polymethylmethacrylate |
| PVA | Polyvinyl alcohol |
| PVD | Physical vapor deposition |
| SLA | Stereolithography |
| SLM | Selective laser melting |
| TEHL | Thermoelastohydrodynamically lubrication |
| TPU | Thermoplastic polyurethanes |
| WAAM | Wire-arc additive manufacturing |

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
