# Peer review of "Surface Properties and Tribological Behavior of Additively Manufactured Components: A Systematic Review"

_lubricants, doi:10.3390/lubricants11060257_

Round 1

Reviewer 1 Report

This review is " Surface properties and tribological behavior of additive manufactured components: a systematic review" related to additively manufactured materials and their tribological properties. I think this review has a novel, and successful literature review and results. However, some points should be revised.

                               I.            The subject is very current and the article is very successful, but ceramic-based coatings can also be made on materials produced by additive manufacturing, which should be added to the literature review section in studies related to them.

                             II.            In addition, metallic materials can be built on top of each other by additive manufacturing. The following suggested studies and relevant paragraphs should be added to them.

·         https://doi.org/10.1016/j.vacuum.2020.109893

·         https://doi.org/10.1007/s42235-021-0055-6

·         DOI 10.1088/2051-672X/accf6c

                          III.            Implant materials are also produced with this innovative method. Coating processes on implant materials should be more detailed.

                         IV.            In one of the chapters, the damage mechanisms (such as wear, corrosion, and tribocorrosion) to which the implant materials are exposed can be explained.

                            V.            In general, the study contains important information. It can be published when the above-mentioned deficiencies are corrected.

Author Response

Thank you very much for the positive feedback on our contribution and the helpful comments. We believe that this will significantly improve our contribution! We have taken your comments into account as follows:

I. Thank you very much for your helpful comment! Indeed, within our review, we found some articles in which ceramic or ceramic-based coatings were created on additively manufactured surfaces, e.g., by oxide formation or classical coating. This is certainly a promising approach. In the first draft manuscript, for example, sources 103, 114, 131, 141. However, since the proportion of papers with classical coatings was quite small compared to the other papers, we did not make a separate subdivision. We have again highlighted the possibility of ceramic coatings at appropriate places to indicate relevance in this area.

II. Thank you for pointing out the exciting contributions. We have included the first two as additionally found contributions in our literature search. Unfortunately, the third article no longer falls within the time period investigated in the analysis, but is very exciting, especially regarding the ceramic coating of additively manufactured components (see Note I.).

III. This aspect is exciting and included in the contributions proposed in II. However, coating processes on implant materials per se do not fall within the focus of our review. Our goal was to provide a broad overview of contributions that make the connection between additive manufacturing and tribology. In our outlook, however, it becomes clear that biomedical applications are highly relevant and are likely to become more important in the future.

IV. Thank you for the helpful comment at this point. It is very useful to highlight that in implant applications (and others as well) corrosive and tribocorrosive aspects often play an important role in addition to pure wear. We had previously only mentioned this aspect in a half-sentence and had now added some more text to emphasize its relevance. We have included this at an appropriate place (line 473) in the manuscript.

V. Thank you very much for your positive assessment of our contribution! We would like to thank you once again for the comments you made earlier, and we believe that our contribution now also addresses these relevant topics more strongly.

Best regards

Reviewer 2 Report

It was a captivating review. I thoroughly enjoyed reading it.

My only observation is that in line 60, it is initially mentioned that the search for works related to AM and tribology was conducted from 2001, but it is later changed to 2011.

Author Response

Dear reviewer,

Thank you for your positive feedback on our article, we were very pleased. Thank you also for pointing out the ambiguous years. During our research, we generally considered all contributions from 2001 onwards, but there were no publications prior to 2011 that fell into the grid with the selected keywords. We have now mentioned this in the manuscript to clarify the procedure once again.

Best regards

Round 2

Reviewer 1 Report

It can be clearly seen that all comments have complately addressed. Thus, this review is stronger and can be accepted for publishing.